RESEARCH                                                                                        Open Access

# DNA polymerase epsilon is required for heterochromatin maintenance in *Arabidopsis*

Pierre Bourguet[1], Leticia López-González[1], Ángeles Gómez-Zambrano[1,2], Thierry Pélissier[1], Amy Hesketh[1], Magdalena E. Potok[3], Marie-Noëlle Pouch-Pélissier[1], Magali Perez[4], Olivier Da Ines[1], David Latrasse[4], Charles I. White[1], Steven E. Jacobsen[3,5], Moussa Benhamed[4] and Olivier Mathieu[1*]

* Correspondence: olivier.mathieu@uca.fr
[1]Institute of Genetics Reproduction and Development (iGReD), Université Clermont Auvergne, CNRS, Inserm, F-63000 Clermont-Ferrand, France
Full list of author information is available at the end of the article

## Abstract

**Background:** Chromatin organizes DNA and regulates its transcriptional activity through epigenetic modifications. Heterochromatic regions of the genome are generally transcriptionally silent, while euchromatin is more prone to transcription. During DNA replication, both genetic information and chromatin modifications must be faithfully passed on to daughter strands. There is evidence that DNA polymerases play a role in transcriptional silencing, but the extent of their contribution and how it relates to heterochromatin maintenance is unclear.

**Results:** We isolate a strong hypomorphic *Arabidopsis thaliana* mutant of the POL2A catalytic subunit of DNA polymerase epsilon and show that POL2A is required to stabilize heterochromatin silencing genome-wide, likely by preventing replicative stress. We reveal that POL2A inhibits DNA methylation and histone H3 lysine 9 methylation. Hence, the release of heterochromatin silencing in POL2A-deficient mutants paradoxically occurs in a chromatin context of increased levels of these two repressive epigenetic marks. At the nuclear level, the POL2A defect is associated with fragmentation of heterochromatin.

**Conclusion:** These results indicate that POL2A is critical to heterochromatin structure and function, and that unhindered replisome progression is required for the faithful propagation of DNA methylation throughout the cell cycle.

**Keywords:** DNA polymerase epsilon, Replication stress, DNA methylation, Heterochromatin, Silencing

## Background

In nearly all eukaryotes, DNA is wrapped around histone proteins to form nucleosomes that allow extensive compaction of the genome while allowing access for important processes such as DNA replication, DNA repair, and transcription. Each nucleosome consists of about 147 bp of DNA wrapped around two molecules of each of four histones H2A, H2B, H3, and H4. Two main states of chromatin organization

can be distinguished in cell nuclei: euchromatin, which contains most genes and is loosely compacted, and heterochromatin, which is enriched in repetitive DNA, gene-poor, and highly compacted. These two main chromatin states associate with distinct patterns of so-called epigenetic marks, namely DNA cytosine methylation and post-translational modification of histone proteins, which influence gene activity in a DNA sequence-independent manner.

In *Arabidopsis thaliana*, pericentromeric heterochromatin contains most of the transposable elements (TEs) of the genome and is associated with high levels of DNA methylation in the three cytosine sequence contexts CG, CHG, and CHH (where H is any base but G). The METHYLTRANSFERASE 1 (MET1) DNA methyltransferase propagates methylation at CG sites upon de novo DNA synthesis during DNA replication, while CHROMOMETHYLASE 3 (CMT3) presumably ensures a similar function at CHG sites. CMT3 is recruited by histone H3 methylation at lysine 9 (H3K9me) deposited by the SU (VAR)3-9 HOMOLOG 4/KRYPTONITE (SUVH4/KYP), SUVH5 and SUVH6 histone methyltransferases, and CHG methylation is in turn needed to recruit SUVH4/5/6 [1]. Whether newly synthesized chromatin during DNA replication is firstly methylated at the DNA level by CMT3 or at the histone H3 level by SUVH4/5/6 is unknown. H3K9me also recruits CHROMOMETHYLASE 2 (CMT2), which is responsible for the maintenance of most genomic asymmetric CHH methylation, and also function partially redundant with CMT3 to methylate CHG sites [2, 3]. It is currently unknown whether CMT2 activity is linked to DNA replication. The remaining fraction of genomic CHH methylation depends on the RNA-directed DNA methylation (RdDM) pathway involving 24-nt small interfering RNAs (siRNAs) and DOMAINS REARRANGED METHYLTRANSFERASE 2 (DRM2), which is responsible for most of de novo DNA methylation. Besides dense DNA methylation, Arabidopsis heterochromatin is additionally enriched in mono-methylation of H3K27 (H3K27me1) deposited by ARABIDOPSIS TRITHORAX RELATED PROTEIN 5 and 6 (ATXR5 and ATXR6), which, like MET1, directly interact with Proliferating Cell Nuclear Antigen (PCNA) and therefore likely function during DNA replication [4, 5]. Finally, the histone H2A variant H2A.W specifically incorporates into Arabidopsis heterochromatin, independently of DNA and H3K9 methylation [6].

Analyses of Arabidopsis DNA and histone methyltransferases mutants have demonstrated that epigenetic patterns are instrumental to both heterochromatin organization and function. Among other biological functions, heterochromatin ensures transcriptional repression of TEs and defects in maintaining heterochromatin epigenetic marks lead to the release of TE silencing [7]. Mutants depleted in these marks also exhibit mis-organization of heterochromatin [8]. At the nuclear level, Arabidopsis heterochromatin typically organizes in structures called chromocenters, which appear smaller in *met1* mutant nuclei due to dispersion of pericentromeric sequences away from chromocenters [9]. Decreased H3K27me1 levels in *atxr5 atxr6* mutant nuclei result in extensive remodeling of chromocenters, which then form unique structures of hollow appearance in association with overreplication of heterochromatin [5, 10]. Current data are consistent with a model wherein TE silencing release in *atxr5 atxr6* may conflict with normal heterochromatin replication leading to the production of extra DNA in heterochromatin [11]. However, this effect is unique to H3K27me1 and loss of other heterochromatin silencing marks does not entail DNA overreplication [12].

Both genetic and epigenetic information must be faithfully transmitted to daughter cells during cell divisions. DNA replication involves a large number of proteins required for chromatin disruption, DNA biosynthesis, and chromatin reassembly. Mutations in several DNA replication-related genes have been reported to destabilize silencing of transgenes and selected endogenous loci in Arabidopsis. These mutations include mutations in the replication protein A2A (RPA2A), the DNA replication factor C1 (RFC1), the flap endonuclease 1 (FEN1), the topoisomerase VI subunit MIDGET, the FASCIATA1 (FAS1), and FAS2 components of the Chromatin assembly factor 1 (CAF-1), as well as mutations in subunits of the three replicative DNA polymerases (Pol), Pol alpha (α), delta (δ), and epsilon (ε) [13–21]. Interfering with DNA replication would be expected to lead to improper propagation of epigenetic patterns; however, evidence for components linking DNA replication with epigenetic inheritance is scarce. None of the corresponding mutants harbor reduced DNA methylation levels [13, 14, 16–20, 22], and although decreased levels of H3K9me2 at desilenced loci were reported in Pol α mutants, the depletion in this mark was surprisingly not associated with detectable changes in DNA methylation [14]. Many of these mutants show a reduced level of H3K27me3 at upregulated genes [16, 17, 23–25], but this cannot explain the release of heterochromatic TE silencing since H3K27me3 is largely excluded from constitutive heterochromatin [26]. Similar to plants, mutations in replisome components provoke silencing defects in fission yeast [27–29]. Yet, replication hindrance has emerged as a mechanism of heterochromatin establishment in yeast and human [30, 31]. Therefore, how replisome mutations interfere with the maintenance of epigenetic marks and TE silencing is unclear.

Here, we identified a strong mutant allele of *POL2A* encoding the catalytic subunit of the DNA Pol ε in a screen for mutants defective in transcriptional silencing. We find that POL2A does not promote the accumulation of heterochromatic marks such as H3K27me1, H3K9me2, or H2A.W, but is required for proper aggregation of heterochromatic domains into chromocenters. We show that POL2A both prevents CHG DNA hypermethylation of TEs and controls their silencing genome-wide. Our data reveal a link between DNA replication and CHG methylation as we find that CHG hypermethylation is a feature common to many mutants for replisome factors. Our data highlight the important role of Pol ε in controlling both heterochromatin organization and function.

## Results

### POL2A maintains gene silencing genome-wide

Certain transgenes can spontaneously undergo silencing, which is subsequently maintained by mechanisms identical to those controlling silencing of endogenous TEs and genes. As such, these transgenes represent unique useful tools to genetically dissect silencing pathways. The L5 transgenic locus, which consists of several repeats of the *β-glucuronidase* (*GUS*) gene under control of the CaMV 35S promoter, has spontaneously undergone transcriptional gene silencing in the L5 line, and mutations in many silencing regulators, or various stresses, can reactivate GUS expression [19, 32–35]. In a genetic screen for mutants defective in L5 transgene silencing, we isolated a mutant named *anxious2* (*anx2*) displaying both GUS reactivation and severe developmental

defects (Fig. 1a b). The *anx2* plant phenotype closely resembled that of mutants of the *POL2A* gene, which encodes the catalytic subunit of the DNA Pol ε responsible for most of the leading strand elongation during eukaryotic DNA replication (Additional file 1: Figure S1A) [15, 36, 37]. We found that L5-GUS expression was also reactivated by the previously published *esd7-1* mutation of *POL2A* [36], here renamed *pol2a-10* (see Additional file 2: Table S1 for allele numbers), and allelic tests demonstrated that release of silencing in *anx2* was caused by a mutation in *POL2A* (Additional file 1: Figure S1B-C). We identified a G to A substitution at the nucleotidic position 8707 of *POL2A* in *anx2*, causing an arginine to histidine substitution at amino acid 1063 (Fig. 1c). Consequently, *anx2* was renamed *pol2a-12*. While analyzing *pol2a-12*, we continued screening of the L5 mutant population for L5 silencing suppressors and isolated two additional mutant alleles of *POL2A*: *anx3* (renamed *pol2a-13*) that has a mutation identical to *pol2a-8*, and *anx4* that we renamed *pol2a-14* (Additional file 1: Figure S1D-E, 1C). Interestingly, *pol2a* mutations associated with silencing defects,

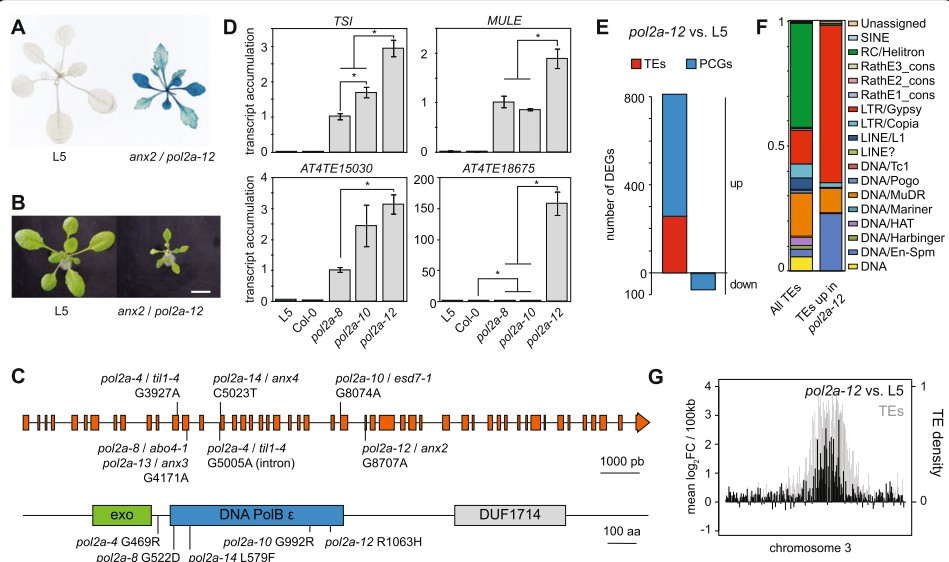

**Fig. 1** Genome-wide release of silencing in a new *pol2a-12* mutant allele. **a** L5-GUS transgene activity detected by X-Gluc histochemical staining in 3-week-old L5 plants and *anx2/pol2a-12* mutants. **b** Representative pictures of 16-day-old L5 and *anx2* plants. Scale bar: 1 cm. **c** (top) Gene model for *POL2A* showing point mutations. The new nomenclature of *pol2a* mutant alleles used in this study is detailed in Table S1. The *til1-4/pol2a-4* mutant allele contains two point mutations in the *POL2A* gene: one in exon 12 and a second one in intron 14. *abo4-1/pol2a-8* is a G to A mutation at position 4171 changing Gly 522 to Asp. The same mutation was identified in *anx3/pol2a-13*. *esd7-1/pol2a-10* is a G to A mutation at position 8074 changing Gly 992 to Arg. *anx4/pol2a-14* is a C to T transition at position 5023 changing Leu 579 to Phe. (bottom) POL2A protein model showing point mutations and their corresponding amino acid changes (positions are indicated relative to the start ATG). Functional domains are indicated according to the conserved domain database from NCBI. DNA PolB ε: DNA polymerase type-B epsilon subfamily catalytic domain. Exo: DNA polymerase family B 3′ to 5′ exonuclease domain. DUF1714: domain of unknown function. **d** Transcript accumulation at four silent loci detected by RT-qPCR, normalized to the *ACTIN2* gene with *pol2a-8* set to 1. Asterisks mark statistically significant differences (unpaired two-sided Student's *t* test, $P < 0.05$). Error bars represent standard error of the mean across three biological replicates. **e** Number of PCGs and TEs detected as differentially expressed in *pol2a-12*. **f** Proportion of TE superfamilies of all TEs in the Arabidopsis genome and of TEs upregulated in *pol2a-12*. **g** Changes in transcript accumulation in *pol2a-12* relative to L5 control plants represented along chromosome 3 by $\log_2$ ratios of average reads per kilobase per million mapped reads (RPKM) over non-overlapping 100 kb bins (black, left *y*-axis). Total TE density is the proportion of TE annotations per 100 kb bins, indicating the pericentromeric region (gray, right *y*-axis)

namely *pol2a-8*, *pol2a-10*, *pol2a-12*, *pol2a-13*, and *pol2a-14*, all lie in the N-terminal part of the POL2A protein carrying the exonuclease and replicative domains (Fig. 1c).

A previous report showed that silencing of a (*35S-NPTII*) transgene and of the endogenous *TRANSCRIPTIONNALLY SILENT INFORMATION* (*TSI*) repeats was released in *pol2a-8* seedlings, and this was shown to occur without changes in DNA methylation [15]. *TSI* transcription was also activated in *pol2a-12*, and of the three *pol2a* mutant alleles analyzed, *pol2a-12* displayed the highest degree of both silencing release and developmental alterations (Fig. 1d, S1A). Analysis of transcript accumulation at other various selected endogenous silent loci confirmed this conclusion and indicated that *POL2A* may play a broader role in controlling silencing genome-wide (Fig. 1d, S1F). To test this, we compared the transcriptomes of *pol2a-12* and WT seedlings generated by RNA sequencing (RNA-seq) and found that almost all (783/860, 90%) differentially expressed loci in *pol2a-12* were upregulated (Fig. 1e). We identified 555 protein-coding genes (PCGs) and 256 TEs upregulated in *pol2a-12*, with upregulated TEs being significantly enriched in LTR/Gypsy retroelements located in pericentromeric heterochromatin (Fig. 1f, g, S1G). Collectively, these data reveal a pivotal role for POL2A in maintaining epigenetic silencing.

### Transcriptional upregulation at genes in *pol2a*

Earlier work indicated that POL2A is involved in transcriptional repression of the *FT* (*FLOWERING LOCUS T*) and *SOC1* (*SUPPRESSOR OF OVEREXPRESSION OF CONSTANS 1*) floral integrator genes by promoting H3K27me3 deposition through a direct interaction with some components of the Polycomb Repressive Complex 2 (PRC2) [24, 36, 38]. To assess the importance of H3K27me3 in POL2A-mediated silencing at the genome-wide scale, we compared H3K27me3 profiles in *pol2a-12* and WT seedlings using chromatin-immunoprecipitation followed by deep sequencing (ChIP-seq). Consistent with previous observations in the *pol2a-8* and *pol2a-10* alleles [36, 38], we found that several flowering genes were transcriptionally upregulated and showed slightly reduced levels of H3K27me3 in *pol2a-12* compared with the WT (Additional file 1: Figure S2A). Noticeably, we found that 249 out of the 555 PCGs upregulated in *pol2a-12* were associated with H3K27me3 in the WT, and these PCGs showed slightly decreased levels of H3K27me3 in *pol2a-12* (Additional file 1: Figure S2B-C). Transcript accumulation from this set of genes was significantly upregulated in a double mutant for the CURLY LEAF and SWINGER H3K27me3 methyltransferases (Additional file 1: Figure S2D) [39], suggesting that H3K27me3 represses their expression. This extends previous observations at the *SOC1* and *FT* genes [24] and suggests that impaired PRC2-mediated H3K27me3 deposition might contribute to about half of PCG upregulation in *pol2a-12*. PCGs upregulated in *pol2a-12* were mostly depleted in H3K4me3 (Additional file 1: Figure S2E) indicating that POL2A does not repress genes associated with H3K27me3/H3K4me3 bivalent chromatin [40]. The *pol2a-8* mutants show increased expression of DNA repair genes, which results from a state of constitutive replication stress [15, 38, 41]. We found that DNA repair genes are also upregulated in *pol2a-12*, and noticeably, they are not marked by H3K27me3 in the WT (Additional file 1: Figure S2F). This suggests that these genes are not controlled by H3K27me3 and that their upregulation in *pol2a-12* is likely triggered by constitutive replicative stress. Supporting

this notion, 35% of *pol2a-12* upregulated PCGs were similarly upregulated in *atxr5/6* mutants (Additional file 1: Figure S2G), which undergo DNA overreplication and DNA damage [10–12, 42], and most of these genes (69%) were not associated with H3K27me3 (Additional file 1: Figure S2G). A last class of *pol2a-12* upregulated PCGs (174), neither marked by H3K27me3 nor upregulated in *atxr5/6*, was enriched for genes involved in biological processes related to cell proliferation, cell cycle, and homologous recombination (24.7% of 174 PCGs) (Additional file 2: Table S2). Elevated rates of homologous recombination and increased S-phase length were reported in *pol2a-8* mutants [15, 41]. Together, our findings suggest that decreased H3K27me3 levels at some loci, constitutive replicative stress, and disturbed cell cycle progression each likely contribute to PCG upregulation in *pol2a-12*.

### POL2A is required for *atxr5/6*-induced heterochromatin overreplication but likely regulates TE silencing independently of H3K27me1

Out of the 256 TEs upregulated in *pol2a-12*, only 15 were associated with H3K27me3 in the WT (Additional file 1: Figure S3A). TEs derepressed in *pol2a-12* were mostly located in pericentromeric heterochromatin (Fig. 1f), which is largely depleted in H3K27me3 but enriched in H3K27me1, a repressive histone modification mediated by ATXR5 and ATXR6 [5, 26, 42]. Similar to *pol2a-12*, transcriptome analysis of *atxr5/6* revealed that up-regulated TEs were strongly enriched for elements belonging to the LTR/Gypsy superfamily (Additional file 1: Figure S3B, [12]). Although *atxr5/6* activated a higher number of TEs than *pol2a-12*, most (87.9%) *pol2a-12* upregulated TEs were also activated in *atxr5/6* (Fig. 2a). This prompted us to investigate whether *pol2a-12* affects H3K27me1 levels. We determined genome-wide H3K27me1 profiles in *pol2a-12* and WT control seedlings using ChIP-seq and compared these with available data for *atxr5/6* mutants [43]. Changes in H3K27me1 in *pol2a-12* were very modest compared with the marked reduction in *atxr5/6* (Fig. 2b, c, S3C), although TEs reactivated in both mutants accumulated similar transcript levels (Fig. 2d). Decreased H3K27me1 level in *atxr5/6* mutants is associated with over-replication of heterochromatic DNA [42] and flow cytometry analyses did not reveal such genome instability in *pol2a-12* mutants (Additional file 1: Figure S3D). Interestingly, combining *pol2a-12* and *atxr5/6* by crossing, we found that the *pol2a-12* mutation strongly suppressed the production of *atxr5/6*-induced extra DNA (Additional file 1: Figure S3D). Heterochromatin over-replication in *atxr5/6* correlates with the appearance of hollow chromocenters [10] and we found that these were also suppressed in *pol2a atxr5/6* (Additional file 1: Figure S3E). Mutations in several genes have been shown to suppress the production of extra DNA in *atxr5/6*. In these mutants, *atxr5/6*-induced TE transcription and overexpression of DNA damage-induced HR genes were also reduced or suppressed [11]. Transcriptome analysis indicated that suppression of heterochromatic DNA over-replication in *pol2a atxr5/6* triple mutants was not accompanied by significant changes in transcript accumulation from *atxr5/6*-reactivated TEs nor by suppression of HR gene overexpression (Fig. 2d, S3F-G). This suggests that *atxr5/6* mutations may affect transcription and replication independently. In support of this interpretation, mutations in the MET1 or CMT3 DNA methyltransferases suppress DNA over-replication but enhance TE derepression in *atxr5/6* mutants [12]. Altogether, our data indicate that POL2A is

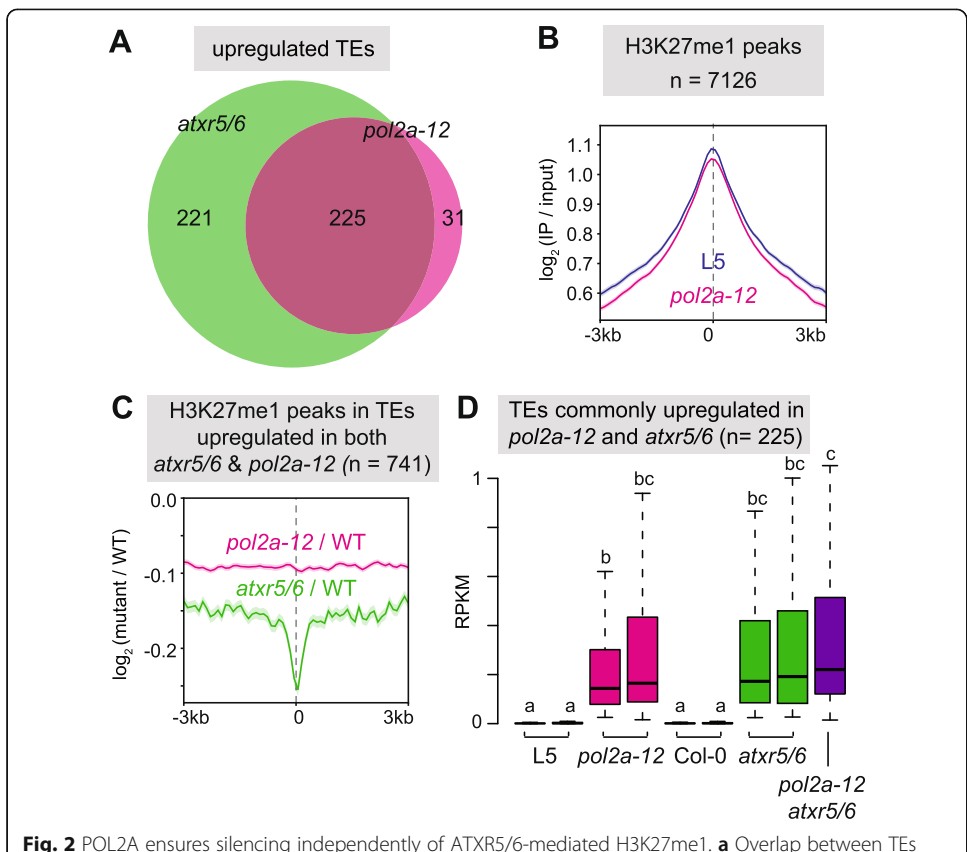

**Fig. 2** POL2A ensures silencing independently of ATXR5/6-mediated H3K27me1. **a** Overlap between TEs upregulated in *atxr5/6* (data from Ikeda et al. [35]) and *pol2a-12*. **b** Metaplots showing average H3K27me1 enrichment (log$_2$ signal over input) at H3K27me1 peaks. Shaded areas show standard deviation. One biological replicate is shown. **c** Metaplots showing H3K27me1 changes (log$_2$ mutant / WT) *pol2a-12* and *atxr5/6* (data from Ma et al. [43]) at H3K27me1 peaks overlapping TEs upregulated in both *atxr5/6* and *pol2a-12*, represented as in **b**. Average of two replicates is shown. **d** Transcript accumulation in reads per kilobase per million mapped reads (RPKM) in indicated genotypes. The effect of genotype was verified with a Kruskal-Wallis rank-sum test. Significant differences between groups, evaluated by a Dwass-Steel-Crichtlow-Fligner test, are indicated by lowercase letters (*P* < 0.05). Two biological replicates are shown for each genotype, except for *pol2a atxr5/6* where only one sample was analyzed

required for *atxr5/6*-induced heterochromatin overreplication but regulates TE silencing largely independently of ATXR5/6-mediated H3K27me1.

## POL2A is required for proper heterochromatin organization independently of H2A.W

In DAPI-stained WT Arabidopsis nuclei, heterochromatin is visualized as large densely stained foci called chromocenters. Nuclei of *pol2a-12* mutants showed visibly reduced heterochromatin content, associated with DAPI-stained foci that were typically smaller and more numerous than WT chromocenters (Fig. 3a, b). WT chromocenters contain highly repeated DNA sequences, including *180-bp* satellite repeats and *45S* rDNA, which were transcriptionally derepressed in *pol2a-12* (Additional file 1: Figure S1F, S4A). Fluorescence in-situ hybridization using probes corresponding to *180-bp* and *45S* rDNA repeats revealed that these repeats were included in the small DAPI-stained foci and were not dispersed throughout *pol2a-12* nuclei (Additional file 1: Figure S4B-C). Immunocytology analyses further indicated that DAPI-stained foci in *pol2a-12* were

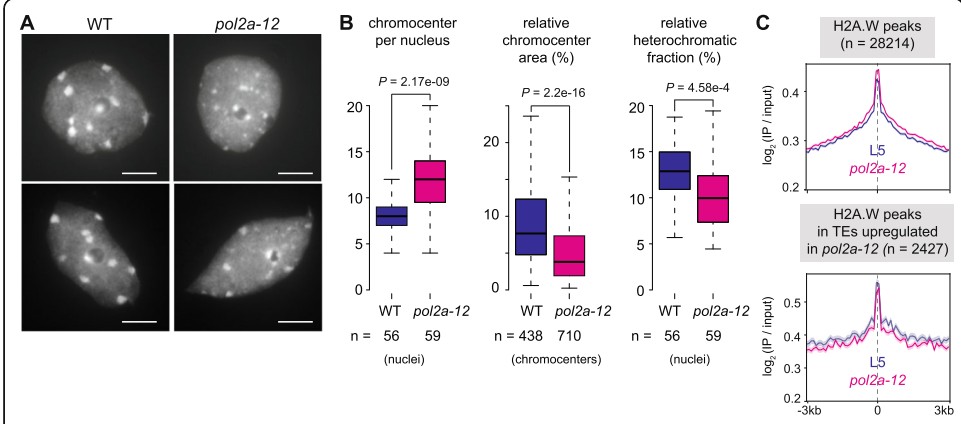

**Fig. 3** Heterochromatin fragmentation in *pol2a-12*. **a** DAPI-stained nuclei extracted from WT and *pol2a-12* plants. Scale bar: 5 μm. **b** Number of chromocenter per nucleus (left), area of individual chromocenter normalized to the entire nucleus area (middle) and relative heterochromatic fraction (right) in WT and *pol2a-12* quantified on 56 and 59 DAPI-stained nuclei, respectively. *P*-values from an unpaired two-sided Student's t-test are indicated. **c** Metaplots showing H2A.W enrichment (log$_2$ signal over input) in L5 and *pol2a-12* at H2A.W peaks (top) and at peaks overlapping TEs upregulated in *pol2a-12* (bottom). Shaded areas show standard deviation. TE annotations were extended 1 kb upstream

associated with H3K27me1 and H3K9me2 like WT chromocenters, suggesting that they still retain heterochromatin features (Additional file 1: Figure S4D). Therefore, we conclude that the small DAPI-stained foci in *pol2a-12* nuclei likely represent dispersed fragments of normally larger WT chromocenters, indicating that POL2A is required for proper higher-order, heterochromatin organization.

The H2A.W histone H2A variant is specifically enriched in Arabidopsis heterochromatin and promotes long-range interactions of chromatin fibers [6, 44]. We quantitatively profiled H2A.W in *pol2a-12* and WT seedlings using ChIP-seq and found that H2A.W levels were largely preserved in *pol2a-12* (Fig. 3c), indicating that defective chromocenter organization in *pol2a-12* is independent of H2A.W incorporation into heterochromatin.

### POL2A prevents DNA hypermethylation of heterochromatin

DNA methylation plays a crucial role in maintaining both heterochromatin structure and silencing. Despite a previous study reporting no change in DNA methylation in *pol2a-8* using methylation-sensitive restriction enzyme assays [15], we used whole genome bisulfite sequencing (BS-seq) to determine genome-wide cytosine methylation profiles in *pol2a-8*, *pol2a-10*, and *pol2a-12*. Surprisingly, average genomic methylation rates were markedly increased at CHG sites in *pol2a* mutants in comparison to the WT (Fig. 4a, S5A). CHG sites consist of three different subcontexts (CAG, CTG, and CCG) [45], which were all hypermethylated in *pol2a-12* (Additional file 1: Figure S5B). Gain in CHG methylation was most prominent at pericentromeric heterochromatin, where we also detected a modest increase in methylation at CHH sites (Fig. 4b). Accordingly, average CHG and CHH methylation rates were increased at TEs in *pol2a* mutants, while CG methylation was largely unaltered at TEs and PCGs (Additional file 1: Figure S5C). Separating TEs based on their chromosomal location further revealed that pericentromeric TEs were strongly hypermethylated at CHG sites and to a lesser extent

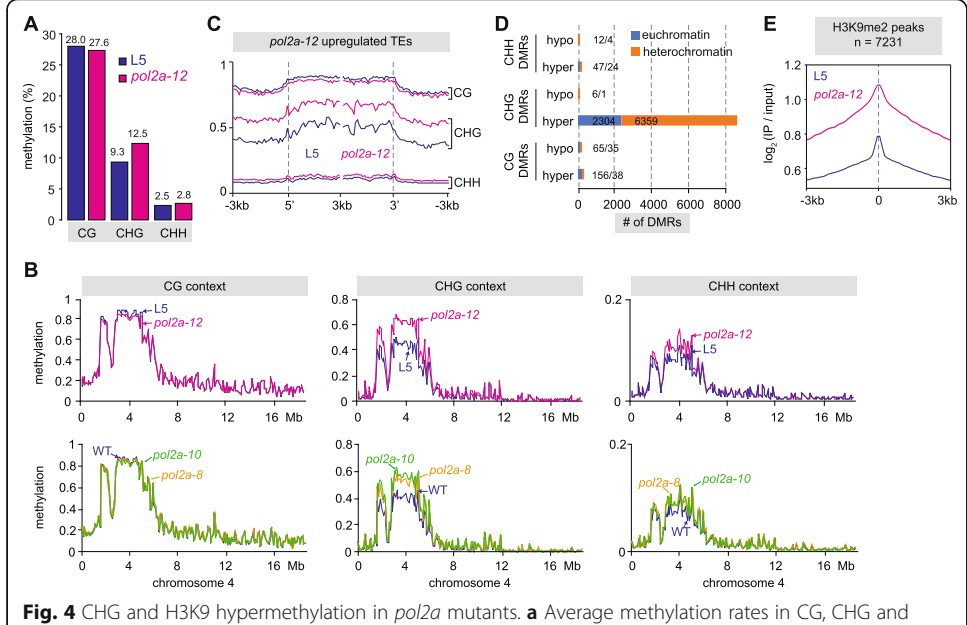

**Fig. 4** CHG and H3K9 hypermethylation in *pol2a* mutants. **a** Average methylation rates in CG, CHG and CHH contexts in L5 and *pol2a-12*. **b** Methylation rates in CG, CHG, and CHH contexts in L5, *pol2a-12*, Col-0 (WT), *pol2a-8*, and *pol2a-10*, averaged over non-overlapping 100 kb bins on chromosome 4. **c** Metaplots showing methylation levels at TEs upregulated in *pol2a-12*. Annotations were aligned to their 5′ or 3′ end and average methylation was calculated for each 100-bp bin from 3 kb upstream to 3 kb downstream. **d** Differentially methylated regions (DMRs) identified in *pol2a-12* (see the "Methods" section). DMRs were further sorted between euchromatin and heterochromatin based on their genomic location. **e** Metaplots showing H3K9me2 enrichment (log$_2$ signal over input) in L5 and *pol2a-12* at H3K9me2 peaks. Shaded areas show standard deviation. One replicate is shown

at CHH positions in *pol2a* mutants, while TEs located on chromosome arms only gained methylation at CHG sites (Additional file 1: Figure S5D). TEs upregulated in *pol2a-12*, which are predominantly located in pericentromeric heterochromatin, gain methylation at both CHG and CHH sites (Fig. 4c, S5E). To further characterize methylation changes in *pol2a*, we determined positions (DMPs) and regions (DMRs) differentially methylated in *pol2a-12* relative to WT. We mostly detected CHG-hypermethylated (CHG-hyper) DMPs and DMRs, which were largely clustered in heterochromatin (Fig. 4d, S5F-G). Only a few CHH hypermethylated positions were detected under the threshold conditions we applied, indicating the most prominent impact of *pol2a* on DNA methylation is increased methylation at CHG sites. TE annotations overlapping CHG-hyper DMRs were strongly skewed for TEs belonging to the LTR/Gypsy superfamily (Additional file 1: Figure S5H), as were those of *pol2a-12* upregulated TEs (Additional file 1: Figure S1G). Analyzing WT methylation rates at *pol2a-12* CHG-hyper DMPs showed that CHG hypermethylation in *pol2a-12* does not occur de novo, but rather targets cytosines already methylated in the WT (median WT methylation rate of 0.32) (Additional file 1: Figure S5I).

Given that pathways maintaining CHG methylation and H3K9 methylation are tightly interwoven [1], increased CHG methylation in *pol2a-12* prompted us to examine H3K9me2 patterns. Using ChIP-seq we found a stark increase in H3K9me2 level in *pol2a-12* at regions already associated with H3K9me2 in the WT, located either in pericentromeres or along chromosome arms (Fig. 4e, S5J). Regions with increased CHG methylation and TEs upregulated in *pol2a-12* also showed H3K9me2 enrichment

(Additional file 1: Figure S5J). Therefore, *pol2a-12* shows genome-wide overaccumulation of H3K9me2 that closely follows CHG hypermethylation. Collectively, these findings demonstrate that POL2A is required for maintaining proper patterns of non-CG methylation and/or H3K9me2. They also reveal that transcriptional derepression of TEs and disruption of heterochromatin organization can, somewhat counterintuitively, occur in a context of increased levels of these two repressive epigenetic marks.

### POL2A and FAS2 influence heterochromatin silencing, organization, and DNA methylation through at least partly distinct pathways

FAS2, together with FAS1 and MSI1 form the CAF-1 complex that incorporates H3.1-associated nucleosomes during DNA replication [46]. In addition to resemblances in their developmental phenotypes (Additional file 1: Figure S6A), *fas2* and *pol2a* mutants exhibit notable similarities in their molecular phenotype. Heterochromatic DNA, in particular LTR/Gypsy TEs, show CHG hypermethylation in *fas2* mutants [22, 47], resembling *pol2a* (Additional file 1: Figure S5H). Furthermore, *fas2* mutants show silencing defects at some genomic loci and FAS2 is required for *atxr5/6*-induced heterochromatin over-replication [18, 42, 48]. We sought to analyze epistasis between *pol2a* and *fas2* mutations; however, we were unable to recover *pol2a-12 fas2-4* double mutants in the progeny of *pol2a-12/+ fas2-4/+* double heterozygotes or either sesquimutant, suggesting a lethal genetic interaction between *pol2a-12* and *fas2-4*. We compared DNA methylation patterns in *pol2a-12* and *fas2-4* and found that in stark contrast with *pol2a*, heterochromatic DNA is hypermethylated not only at CHG sites but also at CG sequence contexts in *fas2-4* (Fig. 4, S5C, S6B), supporting earlier observations [22, 47]. Morever, CHG hypermethylation in *fas2* was much less pronounced at CCG trinucleotides than at CAG and CTG, while CHG hypermethylation in *pol2a* equally affects all three CHG subcontexts (Additional file 1: Figure S6C). Transcriptome analyses using RNA-seq also highlighted differences between *pol2a* and *fas2* mutants. We identified 109 TEs upregulated in *fas2-4*, of which more than half (51.4%) remain efficiently silenced in *pol2a-12* (Additional file 1: Figure S6D). In addition, only 25.6% of the 843 PCGs upregulated in *fas2* accumulated more transcripts in *pol2a* mutants (Additional file 1: Figure S6D). Thus, POL2A and FAS2 regulate transcriptional activity of both common and distinct sets of TEs and PCGs. Finally, *pol2a-12* and *fas2-4* appear to differentially impact nuclear phenotypes. Indeed, although heterochromatin fraction and chromocenter size decreased in *fas2-4* (Additional file 1: Figure S6E-F), there was no significant increase in the number of chromocenters per nucleus, differing from *pol2a* (Fig. 3a-b, Additional file 1: Figure S6E). Additionally, while *fas2* mutants show an increased proportion of endoreduplicated nuclei [49], we did not detect endoreduplication defects in *pol2a-12* (Additional file 1: Figure S3D). Altogether, these differences between *pol2a* and *fas2* molecular phenotypes suggest that POL2A and FAS2 stabilize heterochromatin silencing, heterochromatin organization and prevent DNA hypermethylation through, at least partly, separate pathways.

### DNA methylation changes in *pol2a* are independent of 24-nt siRNA hyper-accumulation

In plants, small interfering RNAs (siRNAs) contribute to the establishment of DNA methylation and part of its maintenance, and are mostly produced from highly DNA-

methylated genomic regions. In light of the DNA methylation changes occurring in *pol2a* mutants, we determined siRNAs accumulation in *pol2a-10* and *pol2a-12* by RNA sequencing of small RNAs (sRNA-seq). Overall relative proportions of 21-nt and 24-nt sRNAs in *pol2a* mutants were similar to those in the WT (Additional file 1: Figure S7A). Determining regions of differential siRNA accumulation identified more regions of decreased 24-nt siRNA abundance (5375) than regions of 24-nt siRNA over-accumulation (4923) in *pol2a-12*. However, the magnitude of increase, on average 6.92-fold, was higher than the magnitude of loss (3.71-fold) (Additional file 1: Figure S7B). Comparatively, changes in 21-nt siRNA accumulation were neglectable with only 75 regions of over-accumulation and 175 regions of 21-nt siRNA loss. Regions of 24-nt siR-NAs over-accumulation in *pol2a* were largely clustered in pericentromeric heterochromatin (Additional file 1: Figure S7C), where we detected increased levels of DNA methylation and H3K9me2 and most silencing defects in *pol2a*. To further investigate the correlation between DNA hypermethylation and changes in 24-nt siRNA abundance in *pol2a* mutants, we determined DNA methylation changes at regions of differential 24-nt siRNA accumulation. Decreased 24-nt siRNA accumulation correlated with a slight reduction in CHH methylation (Additional file 1: Figure S7D). Hypermethylation of CHH sites was not restricted to regions of increased 24-nt siRNAs abundance, but likewise occurred at their flanking sequences, suggesting that at least part of the modest increase in CHH methylation in *pol2a-12* heterochromatin is caused by a mechanism independent of 24-nt siRNA. Importantly, gain in CHG methylation was not restricted to regions of increased 24-nt siRNA accumulation and also occurred at regions of decreased 24-nt siRNA abundance and in their respective flanking regions (Additional file 1: Figure S7D), indicating that CHG hypermethylation in *pol2a* mutants is unlinked to differential 24-siRNA accumulation. Given the extensive dependency of 24-nt siRNA biogenesis on non-CG methylation and H3K9me2 [3], CHG and H3K9me2 hypermethylation most likely explains increased 24-nt siRNA levels in *pol2a*.

## POL2A represses TEs in synergy with CMT3-mediated methylation

To investigate the contribution of the different Arabidopsis non-CG DNA methyltransferases to hypermethylation of heterochromatic CHG sites in *pol2a* mutants, we generated mutant combinations of *pol2a-12* with *drm1 drm2* (*drm1/2*), *cmt2*, or *cmt3* and determined their methylome together with that of *drm1/2*, *cmt2*, *cmt3*, and WT siblings. We found that CHG methylation profiles of TEs in *pol2a* single mutants and in *pol2a drm1/2* and *pol2a cmt2* mutant combinations were virtually identical (Fig. 5a), indicating marginal or no contribution of DRM1/2 and CMT2 to CHG hypermethylation in *pol2a*. Maintenance of CHG methylation is almost exclusively ensured by CMT3, with a minor contribution of CMT2 [2, 3]. We found that accumulation of CMT3 transcripts, and to a lesser extent SUVH4 transcripts, was increased in *pol2a* mutants (Fig. 5b), and that CHG methylation levels were extensively reduced in *pol2a cmt3* compared with their *pol2a* siblings (Fig. 5a). Thus, increased expression of *CMT3* and/or *SUVH4* likely accounts for the increase in CHG methylation triggered by POL2A deficiency. Interestingly, the CHG methylation level in *pol2a cmt3* was still higher than in *cmt3*, particularly in the internal regions of heterochromatic TEs (Fig. 5a, c), the preferred genomic targets of CMT2 [2, 3]. Additionally, hypermethylation at

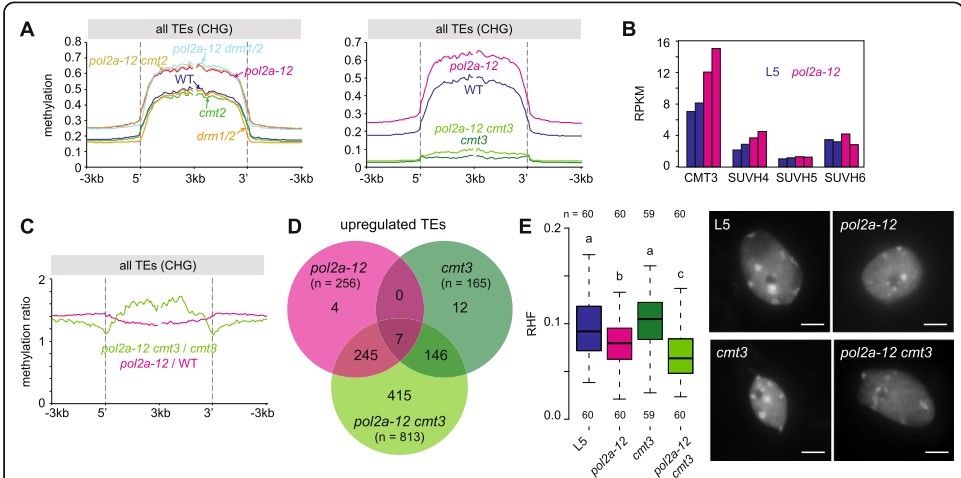

**Fig. 5** CMT3-dependent CHG methylation compensates *pol2a-12* molecular defects. **a** Metaplots showing TE methylation rates in CHG context in the indicated genotypes. Annotations were aligned to their 5′ or 3′ end and average methylation was calculated for each 100-bp bin from 3 kb upstream to 3 kb downstream. **b** Transcript accumulation in reads per kilobase per million mapped reads (RPKM) at the indicated genes. Two replicates per samples are shown. **c** TE methylation changes in CHG context in *pol2a-12* and *pol2a-12 cmt3* normalized to WT and *cmt3*, respectively. **d** Venn diagrams showing the overlap between TEs upregulated in *pol2a-12*, *cmt3* and *pol2a-12 cmt3*. **e** (left) Relative heterochromatic fraction (RHF) evaluated from DAPI-stained pictures of nuclei from the indicated genotypes. The effect of genotype was verified with a Kruskal-Wallis rank-sum test. Significant differences between groups were evaluated by a Dwass-Steel-Crichtlow-Fligner test and are indicated by lowercase letters ($P < 0.05$). The number of analyzed nuclei per genotype is indicated below boxplots. DAPI-stained nuclei extracted from rosette leaves of the indicated genotypes (right). Scale bar: 5 µm

CHH sites, the context favored by CMT2, was more pronounced in *pol2a cmt3* vs. *cmt3* than in *pol2a* vs. WT (Additional file 1: Figure S8A). These findings suggest that CMT2 may take over CMT3 and mediate CHG hypermethylation in the *pol2a cmt3* background.

We used RNA-seq to compare transcriptional changes in *pol2a cmt3* double mutants relative to either single mutants and found that *pol2a* and *cmt3* largely impact TE silencing synergistically (Fig. 5d, S8B-C). Moreover, although *pol2a* and *pol2a cmt3* plants display comparable developmental phenotypes (Additional file 1: Figure S8D), disorganization of heterochromatin was enhanced in *pol2a cmt3* compared with *pol2a* (Fig. 5e). These data suggest the possibility that increased CHG methylation in *pol2a* mutants acts as a compensatory mechanism that counterbalances the release of TE silencing and loss of heterochromatin organization.

## Impairing DNA replication generally triggers CHG DNA hypermethylation and destabilizes silencing

Both ATXR5/6 and POL2A function at DNA replication, and their mutations are associated with constitutive activation of the DNA damage response [10, 15, 41, 42]. We determined the *atxr5/6* methylome in young (13-day-old) seedlings as we did for *pol2a* and found that *atxr5/6* mutants also exhibited increased methylation at CHG sites, although to a lesser extent than in *pol2a* (Fig. 6a), a conclusion that was supported by re-analyzing previously published *atxr5/6* methylome data generated from different tissues (3-week-old rosette leaves) (Additional file 1: Figure S9A). Resembling *pol2a* mutants

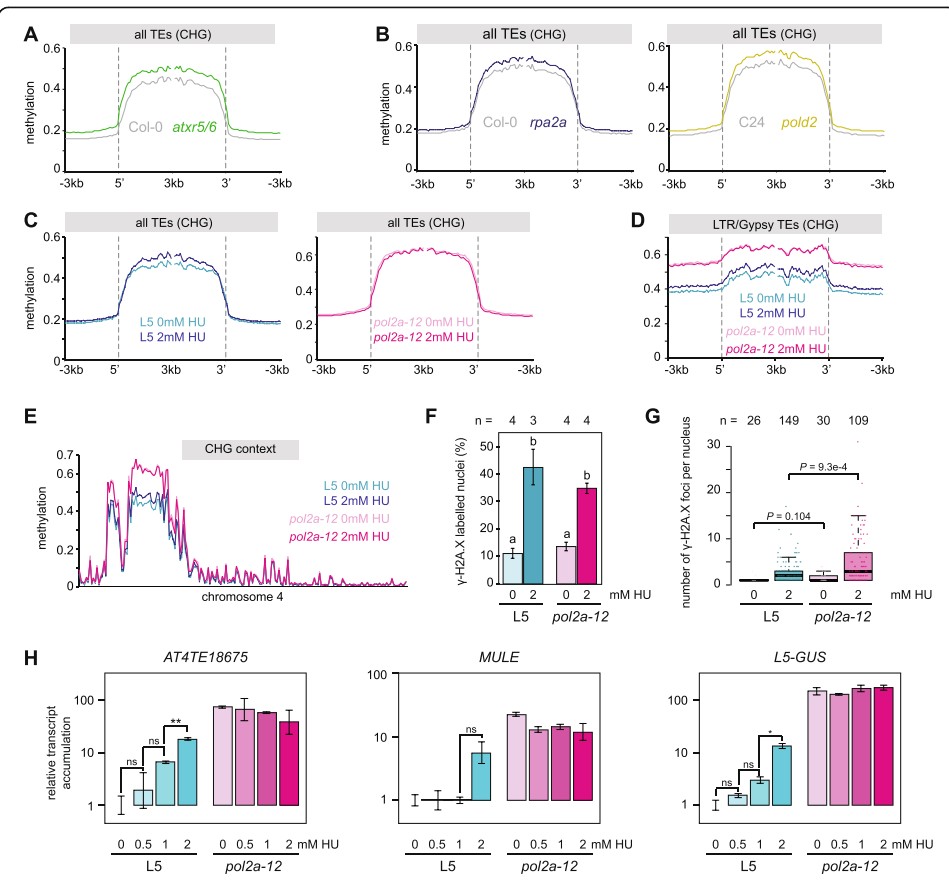

**Fig. 6** DNA replication hindrance provokes CHG hypermethylation and release of silencing. **A**, **B** Metaplots showing TE methylation rates in CHG context in the indicated conditions. Annotations were aligned to their 5′ or 3′ end and average methylation was calculated for each 100-bp bin from 3 kb upstream to 3 kb downstream. For **B**, we used published datasets for *rpa2a* (Stroud et al. [22]) and *pold2* (Zhang et al. [16, 17]). **C**, **D** Average methylation rates in CHG context in L5 and *pol2a-12* in the absence (0 mm) or presence (2 mM) of hydroxyurea (HU) calculated at all TEs (**C**) or LTR/Gypsy TEs (**D**). **E** CHG methylation rates over non-overlapping 100 kb bins on chromosome 4. **F** Proportion of γ-H2A.X labeled nuclei in L5 and *pol2a-12* plants treated or not with HU. A two-way ANOVA showed the significant effect ($P < 2e-16$) of HU treatment. Lowercase letters indicate significant differences between groups using Tukey's post hoc tests ($P < 0.05$). Error bars represent standard error of the mean across three or four biological replicates, as indicated above bars. **G** Number of γ-H2A.X foci per nucleus in L5 and *pol2a-12* plants treated or not with HU, excluding nuclei without γ-H2A.X signal. *P*-values from a two-sided unpaired Wilcoxon rank-sum test are indicated. **H** Transcript accumulation at three silent loci analyzed by RT-qPCR in L5 and *pol2a-12* seedlings treated with various concentrations of HU, normalized to the *ACTIN2* gene with L5 0mM HU set to 1. Asterisks mark statistically significant differences (two-sided unpaired Student's *t* test, \*$P < 0.05$, \*\*$P < 0.005$). Error bars represent standard error of the mean across three biological replicates

(Additional file 1: Figure S5B), DNA hypermethylation in *atxr5/6* was not biased towards a specific CHG subcontext and did not affect CG sites (Additional file 1: Figure S9A-C). Gain in CHG methylation in *atxr5/6* was also associated with increased levels of H3K9me2 (Additional file 1: Figure S9D). We found that *rpa2a* and *pold2*, two other mutants known to affect DNA replication and the DNA damage response [17, 19], also exhibit increased CHG methylation levels (Fig. 6b), while not affecting CG methylation (Additional file 1: Figure S9E). Similarly, we found that TEs gain CHG but not CG methylation in mutants of *MAIL1* (*MAIN-LIKE 1*) (Additional file 1: Figure S9F), which in addition to silencing defects, show constitutive activation of the DNA damage

response and accumulate DNA damage [35, 50, 51]. Interestingly, *pol2a* point mutations associated with CHG hypermethylation all reside in the catalytic replicative domain of POL2A (Fig. 1c). Noticeably, transcription of *CMT3* and *SUVH4/KYP* was unchanged in *atxr5/6*, *pold2*, *mail1*, and *fas2* mutants (Additional file 1: Figure S9G). These findings suggest that DNA replication defects and/or DNA damage generally trigger CHG hypermethylation, independently of CMT3 overexpression.

To test this possibility further, we determined possible DNA methylation changes induced by exposure to hydroxyurea (HU), a drug that causes replication stress by depleting deoxynucleotide triphosphate (dNTP) pools. HU exposure did not alter CG methylation but led to slightly increased methylation levels in all three CHG subcontexts (Fig. 6c, d, S9H-I), preferentially over pericentromeric heterochromatin (Fig. 6e, S9J). HU treatment did not further enhance CHG methylation in *pol2a-12* (Fig. 6c–e, S9H-J), and regions with increased CHG methylation in *pol2a-12* also tended to gain CHG methylation in HU-treated WT plants (Fig. 6d, S9K). Like in *atxr5/6*, *pold2*, and *mail1* mutants, HU-induced CHG hypermethylation was not associated with upregulated CMT3 transcript levels (Additional file 1: Figure S9L). These data suggest that replicative stress and/or DNA damage induce CHG hypermethylation without altering CMT3 expression levels. This mechanism is likely at play in *pol2a* mutants, which exhibit exacerbated CHG hypermethylation as a result of concomitant *CMT3* overexpression.

High concentrations of HU lead to DNA damage and *atxr5/6*, *pold2*, and *mail1* also accumulate DNA damage [10, 50–53]. To try and discriminate whether CHG hypermethylation results from replicative stress or from accumulation of DNA damage, we used immunocytology to detect phosphorylated H2A.X (γ-H2A.X). In response to DNA damage, the histone variant H2A.X becomes rapidly phosphorylated at sites of DNA breaks and detection of discrete γ-H2A.X foci can be used as a proxy to monitor double-strand break formation [54]. We found that *pol2a-12* mutant nuclei did not detectably accumulate more γ-H2A.X foci than WT nuclei under normal growth conditions (Fig. 6f, g). Our HU treatment conditions induced DNA damage in both WT and *pol2a-12* (Fig. 6f, g), and this effect was more pronounced in *pol2a-12* (Fig. 6g). Importantly, HU-induced DNA damage in *pol2a* mutants did not correlate with an increase in CHG methylation in the treated mutant (Fig. 6c–e). These data suggest that CHG hypermethylation in *pol2a-12* may primarily be triggered by replicative stress and not by DNA damage.

We also tested whether HU treatment may destabilize silencing and found that transcript accumulation at selected silent loci was increased upon exposure of WT plants to HU (Fig. 6h). Additionally, HU treatment did not significantly modify silencing release at these loci in *pol2a-12* (Fig. 6h). These findings suggest that HU exposure and POL2A mutations disturb silencing through a common pathway, supporting that constitutive replication stress contributes to silencing defects in *pol2a* mutants.

## Discussion

Contrasting with earlier conclusions [15], our genome-wide methylation analyses showed that mutations of *POL2A* impact DNA methylation profiles and lead to a sharp increase in CHG methylation. This hypermethylation is not associated with significant changes in genomic profiles of H2A.W and H3K27me1, two hallmarks of Arabidopsis

heterochromatin. Consistent with the fact that the mechanisms maintaining CHG methylation and H3K9 methylation are tightly linked, CHG hypermethylation is associated with increased levels of H3K9me2 in *pol2a* mutants. We found that mutations in other DNA replication-related genes, including *RPA2A*, *POLD2*, and *ATXR5/6*, previously thought not to alter DNA methylation [5, 17, 22], are also associated with increased CHG methylation to various degrees, and at least in *atxr5/6*, with increased H3K9me2 levels. The gain in CHG methylation is stronger in *pol2a-12*, very likely because *CMT3* expression is enhanced in this mutant, although what causes *CMT3* overexpression in *pol2a-12* remains to be elucidated. RPA2A, ATXR5/6, POLD2 (Pol δ), and POL2A (Pol ε) function in coordination with, or at the core of, the replisome and *pol2a* mutations associated with CHG hypermethylation reside in the replicative domain of POL2A. We also detected hypermethylation of CHG sites in plants treated with HU, which causes replication stress by depleting cellular dNTP pools. HU-induced replication stress activates the S-phase checkpoint, resembling *pol2a* mutants where it is constitutively activated [41, 55]. Interestingly, exposing *pol2a* plants to HU did not dramatize CHG hypermethylation, suggesting that *pol2a* mutations and HU provoke hypermethylation through a common pathway. Furthermore, we found no evidence of increased DSB accumulation in *pol2a-12*. Altogether this suggests that replication stress is a trigger for increased CHG and/or H3K9 methylation.

FAS2 is a subunit of the CAF-1 complex, which incorporates H3.1 during DNA replication. H3.1 mostly occupies pericentromeric heterochromatin in differentiated cells, while it is replaced by the H3.3 variant at genes in a transcription-dependent manner [56–58]. H3.3 favors gene body methylation, likely by preventing recruitment of H1 that inhibits DNA methylation by restricting the access of the DNA methyltransferases MET1, CMT2, and CMT3 to DNA [2, 59, 60]. In *fas2* mutants, replacement of H3.1 by H3.3 and/or decrease in H1, correlate with a global increase of heterochromatic DNA methylation in all cytosine sequence contexts [22, 47, 61, 62]. Because CG methylation remains largely unaltered in *pol2a*, *atxr5/6*, *rpa2a*, *pold2*, and *mail1* mutants or after HU exposure (Fig. 4, S5, S9), DNA hypermethylation unlikely results from perturbed genomic distribution of H3.1, H3.3, or H1 in these backgrounds. Recruitement of CMT3 at CAG and CTG sites predominantly relies on H3K9 methylation mediated by SUVH4/KYP, whereas the redundant activities of SUVH5 and SUVH6 are required to target CMT3 at CCG sites [45]. Interestingly, we found that CHG hypermethylation is biased towards CAG and CTG contexts in *fas2* mutants, suggesting that SUVH4 might have a preference for H3.3 over H3.1 and/or might be antagonized by H1.

CHG methylation is likely maintained shortly after the passage of the replication fork as CMT3 is exclusively associated with H3.1 in vivo and is highly expressed in replicating cells [63]. Increased CHG methylation in *pol2a* mutants and HU-treated plants correlates with S-phase checkpoint activation [15, 41, 64], upon which DNA replication is halted until checkpoint-dependent pathways restore cellular conditions suitable for replisome progression [55]. We propose that replication arrest provides CMT3 and/or SUVH4/5/6 with a wider time-window to accomplish their enzymatic activities, resulting in more efficient maintenance and thus increased levels of CHG and H3K9 methylation. Interestingly, replication arrest in human cells leads to an accumulation of chaperone-bound histones marked with H3K9 methylation, which are rapidly incorporated upon resumption of DNA replication [65]. Given the tight link between CHG and

H3K9 methylation in plants, a similar mechanism may explain the prevalence of CHG hypermethylation in the context of constitutive replication stress in Arabidopsis.

DNA transposons are mobilized during DNA replication, and LTR retrotransposons are preferentially inserted at sites of replication fork arrest [66]. In that regard, fork arrest and slower S-phase completion caused by replication stress and checkpoint activation are likely to favor TE mobilization. Increased CHG and H3K9 methylation, by limiting the release of TE silencing and maintaining heterochromatin organization (Fig. 5), may have evolved as a mechanism safeguarding genome integrity in cells undergoing replication stress.

Several studies reported TE silencing defects in replisome-related mutants and in plants treated with DNA-damaging agents, where replication is expectedly disturbed [5, 13–20]. We show that HU can alleviate TE silencing but does not enhance *pol2a*-induced release of silencing (Fig. 6h), which points to replicative stress as a cause of loss of TE silencing in *pol2a* mutants. Chromocenter organization is drastically altered in *pol2a*, which also displays the release of silencing albeit increased levels of CHG and H3K9 methylation. Analyses of *pol2a cmt3* double mutants indicate that DNA hypermethylation compensates silencing release and loss of chromocenter organization. Mutants showing impaired heterochromatin organization, including *met1*, *ddm1*, *atxr5/6*, and *mail1*, exhibit silencing defects [5, 9, 35, 67, 68]. Chromocenter disruption in mutants lacking histone H1 is associated with only weak derepression of few TEs [69]; however, loss of H1 induces increased heterochromatin methylation at CG and CHG sites [2, 69], which likely counterbalances silencing release. Therefore, although the extent of its contribution remains difficult to evaluate, it is tempting to speculate that loss of higher-order heterochromatin organization participates in destabilizing silencing in *pol2a* mutants.

## Conclusions

Our study demonstrates that Pol ε is essential for preserving both heterochromatin structure and function by enforcing chromocenter formation and TE silencing. Furthermore, it reveals that proper DNA replication generally prevents the appearance of aberrant DNA methylation patterns.

## Methods

### Plant material

Plants were grown in soil in long-day conditions (16 h light, 8 h dark) at 23 °C with 60% relative humidity. The *atxr5 atxr6* (SALK_130607, SAIL_240_H01), *cmt3-11* (SALK_148381), *cmt2-3* (SALK_012874C), *drm1-2 drm2-2* (SALK_031705, SALK_150863), *fas2-4* (SALK_033228), and *pold1* (also named *gis5*) [25] mutant lines used in this study were all in a Col-0 genetic background. The *esd7-1* mutant allele, originally isolated in Ler-0, was repeatedly backcrossed in Col-0 [36] while *abo4-1* was generated in a Col-0 background carrying a *glabra1* mutation [15]. The *anx2* (*pol2a-12*), *anx3* (*pol2a-13*), and *anx4* (*pol2a-14*) mutant alleles reported in this study were isolated from a population of mutagenized L5 plants that we previously described [35]. The *pol2a-12* mutant was backcrossed once to the L5 line before analysis. Its developmental phenotype was stable through six backcrosses.

For genome-wide profiling of *pol2a-12 cmt3-11* double mutants and controls, all plants were derived from a F1 parent obtained by crossing *pol2a-12* (3rd backcross) and *cmt3-11*. Siblings were genotyped for *pol2a-12* and *cmt3-11* mutations. For methylome studies, pools consisted of 17 WT plants, 58 *pol2a-12* single mutants, 17 *cmt3-11* single mutants, and 43 *pol2a-12 cmt3-11* double mutants. The same procedure was followed to generate methylomes of *pol2a-12 cmt2-3* double mutants, from a cross of *pol2a-12* (1st backcross) with *cmt2-3*. Sixteen plants were pooled for *cmt2-3*, 19 for *pol2a-12 cmt2*. For *pol2a-12 drm1/2* methylome, two F2 *pol2a-12/+ drm1/2* plants were isolated from a cross between *pol2a-12* (1st backcross) and *drm1/2*. Their F3 progeny was genotyped to pool 15 plants for *drm1/2* and 27 plants for *pol2a-12 drm1/2*.

### Histochemical staining

Whole seedlings or rosette leaves were vaccum inlfiltrated twice 5 min with 3 ml of X-Gluc staining solution (50 mM $Na_xH_xPO4$ pH 7; 10 mM EDTA; 0.2% Triton-X-100; 0.04% X-Gluc) and incubated 24 h at 37 °C in the dark. Chlorophyll was subsequently cleared with repeated washes in ethanol at room temperature.

### Transcript analysis

About 30–40 mg of fresh tissues were used for total RNA extraction with TRI Reagent (Sigma), following the manufacturer's instructions. RNA (8 μg) was treated with 12 units of RQ1 DNase (Promega) for 1 h at 37 °C and further purified by phenol-chloroform extraction and ethanol precipitation. One-step reverse-transcription quantitative PCR (RT-qPCR) was performed starting from 50 ng of RNA using the Sensi-FAST™ SYBR® No-ROX One-Step kit (Bioline) on an Eco™ Real-Time PCR System (Ilumina), following a program of 10 min at 45 °C, 5 min at 95 °C, 40 cycles of 20 s at 95 °C, and 30 s at 60 °C. Amplification specificity was evaluated by analyzing a melting curve generated at the end of the reaction. Amplification of the *ACTIN2* gene transcripts was used as a reference for normalization and data were analyzed according to the $2^{-\Delta\Delta Ct}$ method. End-point RT-PCR was performed using the one-step RT-PCR kit (QIAGEN) following the manufacturer's instructions, in a final volume of 10 μl starting from 50 ng of RNA. Primers used in this study are described in Additional file 2: Table S3.

### mRNA sequencing

Total RNA was extracted from 13-day-old seedlings and treated as indicated above. Two biological replicates were collected for each genotype, except the *pol2a atxr5/6* triple mutant for which we collected one sample (Additional file 2: Table S4 and Additional file 1: Figure S10A). Libraries were prepared using the TruSeq Stranded RNA stranded protocol (Illumina) and sequenced on a HiSeq 2500 instrument (Illumina) at Fasteris S.A. (Geneva, Switzerland) to generate ~ 28–52 M 50-bp single-end reads (see Additional file 2: Table S4 and Additional file 1: Figure S10A). To detect differential expression at protein coding genes (PCGs), we used a pipeline previously described in Bourguet et al. [34]. Only PCGs detected as differentially expressed in both replicates were retained. Gene ontology analysis was performed with PANTHER14.1 Overrepresentation Test (12/03/2019 release) [70]. To detect differentially expressed TEs, reads were aligned to the Arabidopsis TAIR10 genome using STAR version 2.5.3a [71]

retaining multi-mapped reads mapping up to 10 positions. Subsequent read counting was performed with featureCounts version 1.6.0 [72] on the TAIR10 TE annotations. Normalization and differential analyses were done using DESeq2 version 1.14.1 [73] with default parameters. Only loci with Benjamini-Hochberg adjusted $P$ values < 0.05 and with a $\log_2$-fold change $\geq 1$ or $\leq -1$ were considered differentially expressed. TEs with at least 10% of their length overlapping a PCG annotation were excluded from the analysis. RPKM calculation at TEs annotations, in contrast with PCGs, included reads from both strands. We also re-analyzed publicly available data for *atxr5/6, mail1* (ERR1593751-ERR1593754, ERR1593761, ERR1593762 [35]) and *pold2* (GSM2090066-GSM2090071 [16, 17]).

In mRNA-seq and BS-seq (see below) analyses, allocation of genomic features to genomic compartments was based on the chromosomal pericentromeric heterochromatin coordinates previously defined by Bernatavichute et al. [74] based on the distributions of TEs, PCGs, and DNA methylation. Annotations lying within these coordinates were deemed pericentromeric, while annotations overlapping with or located outside these coordinates were assigned to chromosome arms.

### Chromatin-immunoprecipitation followed by sequencing

Histone H3 methylation ChIP-seq was performed on 13-day-old seedlings grown in vitro following a previously described procedure [75] with minor modifications [76]. Two biological replicates were collected for each genotype (see Additional file 2: Table S4 and Additional file 1: Figure S10B). Briefly, tissues were fixed in 1% (v/v) formaldehyde and homogenized in liquid nitrogen. After nuclei isolation and lysis, chromatin was sonicated in a Covaris S220 following the manufacturer's instructions, and shearing efficiency was verified on a gel. Immunoprecipitation was performed with antibodies for H3K27me3 (Millipore 07-448), H3K27me1 (Millipore 07-449), and H3K9me2 (Millipore 07-441). After reverse-crosslink and phenol-chloroform DNA purification, libraries were constructed with the NEBNext® Ultra™ DNA Library Prep Kit for Illumina® (NEB), following the manufacturer's instructions. Sequencing was carried out on a NextSeq 500 instrument to generate ~ 30–106 M 76-bp single-end reads. For H2A.W ChIP-seq, we used a previously described antibody [6]. Libraries were prepared using the NuGEN Ovation Ultra Low System V2 kit from 10-day-old seedlings, according to the manufacturer's instructions, and were sequenced on an Illumina HiSeq 2500 instrument to generate ~ 28–55 M 77-bp single-end reads. Reads were mapped to the Arabidopsis TAIR10 reference genome using STAR version 2.5.3a [71], allowing for two mismatches, and retaining only uniquely mapped reads. Read count was normalized to library size (RPM) and further normalized to the input signal. Peaks were called with MACS2 v2.1.1 [77] using the WT sample with an effective genome size of 9.6E7 and default mfold bounds [5–50] except for H3K27me1 where mfold bounds were broadened [3–50] to allow model building. Narrow peaks were further filtered to retain only peaks with at least two-times more coverage relative to control input DNA or H3. Average metaplots and file manipulations were performed with deepTools v3.1.2 [78] and bedtools v2.26.0 [79].

Publicly available ChIP-seq data for H3, H3K27me1 and H3K9me2 in *atxr5/6* mutants (GSM3040049- GSM3040052, GSM3040059, GSM3040060, GSM3040062, GSM3040063,

GSM3040069-GSM3040072 [43]) were re-analyzed similarly except that read count was normalized to the H3 signal. In Fig. 2c, normalization was restricted to library size to allow comparison of *pol2a-12* and *atxr5/6* data. Replicates were averaged for data representation.

### Nuclei isolation and microscopy

Rosette leaves were fixed in 4% (v/v) formaldehyde 10 mM Tris-HCl for a minimum of 1 h at room temperature, then rinsed in water, and dried and chopped with a razor blade in 150 μl of extraction buffer from the CyStain UV Precise P kit (Partec) in a petri dish. Tissues were passed through a 30-μm filter to isolate nuclei and kept on ice. The procedure was repeated by adding 250 μl of extraction buffer to the petri dish. After 2 min on ice, 10–15 μl of nucleus extract were supplemented with an equal volume of 60% acetic acid on a slide and stirred continuously with fine forceps on a 45 °C metal plate for 3 min, 60% acetic acid was added again and stirred for 3 min. The plate was cleared with an excess of an ethanol/acetic acid solution (3:1), air-dried and mounted with DAPI in Vectashield mounting medium (Vector Laboratories). Nuclei were visualized on a Zeiss Axio Imager Z1 epifluorescence microscope equipped with a PL Apochromat 100X/1.40 oil objective and images were captured with a Zeiss AxioCam MRm camera using the Zeiss ZEN software. The number of DAPI-stained foci and their area relative to that of the entire nucleus was calculated with the ImageJ software to evaluate the number of chromocenters per nucleus and the relative chromocenter area, respectively. The relative heterochromatin fraction was computed for each nucleus by calculating the ratio of the signal intensity at chromocenters over that of the entire nucleus. Immunocytology and fluorescent in situ hybridization were performed as previously described [35].

Immunolocalization of γ-H2A.X was performed as described previously [80]. γ-H2A.X foci were counted by using IMARIS 7.6 software and spot detection method.

### Bisulfite sequencing (BS-seq)

Genomic DNA was extracted from 13-day-old seedlings with the Wizard Genomic DNA Purification Kit (Promega), following the manufacturer's instructions. Sodium bisulfite conversion, library preparation, and sequencing on a Hiseq 2000 or a Hiseq4000 were performed at the Beijing Genomics Institute (Hong Kong) from one microgram of DNA, producing ~ 27–69 M 101-bp paired-end reads (see Additional file 2: Table S4). Our analysis also included publicly available BS-seq datasets for the following mutants: 1st generation *fas2-4* (GSM2800760, GSM2800761 [47]), *pold2* (GSM2090064, GSM2090065 [16, 17]), *atxr5/6* (GSM3038964, GSM3038965, GSM2060541, GSM2060542 [11, 43]), *ddb2-3* (GSM2031992, GSM2031993 [81]), *mail1* (ERR1593765-ERR1593768, [35]), and *rpa2a* and *bru1* (GSM981048, GSM980999, GSM980986 [22]).

Reads were filtered to remove PCR duplicates, using a custom program that considered a read pair duplicated if both reads from a pair were identical to both reads of another read pair. Libraries were mapped to TAIR10 with BS-Seeker2 v2.1.5 [82] using Bowtie2 with 4% mismatches and methylation values were called from uniquely mapped reads. Only cytosines with a minimum coverage of 6 reads were retained. We used a previously described method to detect DMPs and DMRs [35], with minimum

methylation differences after smoothing of 0.4, 0.2, and 0.1 respectively for the CG, CHG, and CHH contexts. To calculate average methylation levels at specific regions, we first determined the methylation rate of individual cytosines and extracted the average methylation rate of all cytosines in the region.

### Flow cytometry

Nuclei extraction and flow cytometry profiling were performed as described in [35].

### sRNA sequencing

Total RNA purified from immature inflorescences was used to generate small RNA libraries (TruSeq small RNA; Illumina), which were sequenced on an Illumina HiSeq 2500 instrument at Fasteris S.A. (Geneva, Switzerland) generating 22–30 M 50-bp single end reads. Reads were post-filtered for 18–26-nt insert size, leaving ~ 12–15 M reads per library (see Additional file 2: Table S4). Reads were mapped to the Arabidopsis TAIR10 genome using TopHat without allowing mismatches. We retained both uniquely mapping reads and multi-mapping reads, the latter being randomly distributed across mapping positions. Read count was normalized to the total amount of 18–26 nucleotide mapping reads within each library.

### Hydroxyurea treatment

Seeds were sterilized 10 min in calcium hypochlorite (0.4%) with 80% ethanol, washed in 100% ethanol, and dried and sowed on solid Murashige and Skoog medium containing 1% sucrose (w/v). After 3 days of stratification, plants were grown 4 days to allow full germination, and subsequently transferred with sterile forceps onto fresh medium supplemented or not with various concentrations of water-dissolved hydroxyurea (Sigma). Seedlings were then grown for 9 days before collection for molecular analysis.

### Statistical analysis

Means and standard errors of the mean were calculated from independent biological samples. All analyses were conducted with R version 3.6.1 [83]. All boxplots had whiskers extend to the furthest data point that is less than 1.5-fold interquartile range from the box (Tukey's definition). Differences in mean for RT-qPCR data were tested using a two-sided unpaired Student's $t$ test with Welch's correction with the t.test function. Kruskal-Wallis rank-sum test was performed with the native kruskal.test R function, and Dwass-Steel-Critchlow-Fligner post hoc tests were made using the pSDCFlig function with the asymptotic method from the NSM3 package [84].

### Supplementary Information

---

**Additional file 1: Figure S1.** Release of transcriptional silencing in three new *pol2a* mutants. **Figure S2.** Contribution of H3K27me3 in POL2A-dependent gene silencing. **Figure S3.** POL2A is required for heterochromatin over-replication in *atxr5/6*. **Figure S4.** DNA repeats, H3K27me1 and H3K9me2 at *pol2a-12* chromocenters. **Figure S5.** DNA methylation and H3K9me2 profiles in *pol2a* mutants. **Figure S6.** Comparison of *pol2a* and *fas2* molecular phenotypes. **Figure S7.** Changes in small RNA accumulation in *pol2a* mutants. **Figure S8.** Characterization of *pol2a cmt3* double mutants. **Figure S9.** DNA methylation profiles in mutant and drug contexts of replicative stress. **Figure S10**. Comparison of sequencing replicates.

---

**Additional file 2: Table S1.** *POL2A* mutant alleles. **Table S2.** Protein-coding genes upregulated in *pol2a-12*, not upregulated in *atxr5/6* and not marked by H3K27me3 ($n = 174$). **Table S3**. List of primers used in this study. **Table S4.** Total read counts and mapping statistics for the sequencing data generated in this study.

**Additional file 3.** Review history.

### Acknowledgements
We thank Yoko Ikeda for help in the initial characterization of *pol2a-12* and Cécile Raynaud (IPS2, Saclay, France) for conceptual input and critical reading of the manuscript.

### Peer review information
Yixin Yao and Kevin Pang were the primary editors of this article and managed its editorial process and peer review in collaboration with the rest of the editorial team.

### Review history
The review history is available as Additional file 3.

### Authors' contributions
Conceptualization: PB TP OM. Data curation: PB TP MEP OM. Formal analysis: PB TP MEP ODI OM. Funding acquisition: CW SEJ MB OM. Investigation: PB TP MNPP AH AGZ LLG MEP MP DL ODI OM. Methodology: PB TP MNPP OM. Project administration: PB OM. Resources: PB TP SEJ MB CW OM. Supervision: OM. Validation: PB TP MNPP MEP OM. Visualization: PB OM. Writing – original draft: PB OM. Writing – review and editing: PB OM. The authors read and approved the final manuscript.

### Funding
This work was supported by CNRS, Inserm, Université Clermont Auvergne, Young Researcher grants from the Auvergne Regional Council (to O.M.), an EMBO Young Investigator award (to O.M.), and a grant from the European Research Council (ERC, I2ST 260742 to O.M.). P.B. was supported by a PhD studentship from the Ministère de l'éducation nationale, de l'enseignement supérieur et de la recherche. Work in the Jacobsen lab was supported by NIH grant R35 GM130272. S.E.J. is an investigator of the Howard Hughes Medical Institute. The funders had no role in study design, data collection and analysis, decision to publish, or preparation of the manuscript.

### Availability of data and materials
The datasets supporting the conclusions of this article are available in the European Nucleotide Archive (ENA; https://www.ebi.ac.uk/arrayexpress/) repository, under accession numbers E-MTAB-8938-40 and E-MTAB-9046 [85].

### Ethics approval and consent to participate
Not applicable.

### Consent for publication
Not applicable.

### Competing interests
The authors declare that they have no competing interests.

### Author details
[1]Institute of Genetics Reproduction and Development (iGReD), Université Clermont Auvergne, CNRS, Inserm, F-63000 Clermont-Ferrand, France. [2]Present Address: Instituto de Bioquímica Vegetal y Fotosíntesis, CSIC-Cartuja, Avda, Américo Vespucio, 49., 41092 Sevilla, Spain. [3]Department of Molecular, Cell and Developmental Biology, University of California, Los Angeles, Los Angeles, CA 90095, USA. [4]Institute of Plant Sciences Paris-Saclay (IPS2), CNRS, INRA, University Paris-Sud, University of Evry, University Paris-Diderot, Sorbonne Paris-Cite, University of Paris-Saclay, Batiment, 630, 91405 Orsay, France. [5]Howard Hughes Medical Institute, University of California, Los Angeles, Los Angeles, CA 90095, USA.

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

## 

