## [**Additional file 3.** Review history. · Genome Biology]

Review History

First round of review

Reviewer 1

Are you able to assess all statistics in the manuscript, including the appropriateness of statistical tests used? No, I do not feel adequately qualified to assess the statistics.

Comments to author:

Bourguet et al. reported the isolation of several mutant alleles of DNA Pol ϵ through genetic screen for mutants that reactivate the GUS expression in Arabidopsis L5 line. They characterized the profiling of H3K27me3, H3K27me1, H3K9me2, DNA methylation, mRNA and small RNA-seq in WT, pol2a and related double or triple mutants as well as under HU treatments. Based on these analysis, the authors try to explore the function of DNA Pol ϵ in heterochromatin maintenance in Arabidopsis. Please see some major concerns on the manuscript.

1. The authors generated a lot of sequencing data in this study, but the details of those datasets are missing, including data quality and correlation between replicates, etc. The authors should have a basic description and a comprehensive table with these details in the manuscript.
2. Line 180-184. The statement "...suggests that about half of PCG upregulation in pol2a-12 likely results from impaired PRC2-mediated H3K27me3 deposition..." might overstate the importance of H3K27me3. How about other types of histone modifications except for H3K27me3? I would rather say "...impaired PRC2-mediated H3K27me3 deposition might contribute to the about half of PCG upregulation in pol2a-12...", or "they are correlated". Similar statement in L197 needs to be considered. In addition, the authors need to provide evidence to support that these upregulated PCG are not randomly overlapped with H3K27me3 mark.
3. Line 218: More data would be required to conclude that "POL2A is required for atrx5/6-induced heterochromatin overreplication", especially with several other negative correlations.
4. The authors try to test the relationship between POL2A and other genes involving heterochromatin silencing across a couple of paragraphs. However, it seems that POL2A exerts its functions independently of these examined genes/proteins, like ATXR5/6, H2A.W and FAS2. These negative results provide limited knowledge for understanding the importance of POL2A as a central coordinator of heterochromatin structure and function. Further, some data, such as sRNA-seq presented, could not well support its role of POL2A in heterochromatin structure and function.

Some minor concerns:

1. A complementation test using genomic fragment containing wild-type DNA Pol ϵ gene to rescue the phenotype would provide solid genetic evidence for the study.
2. According to above words, I think the title of the manuscript overstates the findings and the statement "DNA polymerase epsilon is a central coordinator" is far-fetched.
3. Line 59-60, the authors claims that "euchromatin and heterochromatin are two main different organization states..." is oversimplified and could not match the conclusions from recent published a couple of plants ENCODE papers. At least 4-5 different types of chromatin states were reported.
4. Line 164. Please provide numbers for total DEGs.
5. L187-194: please add the percentages also for the upregulated PCGs involved in replicative stress and cell cycle.
6. Lots of genes are not given a full name at the first time appearance, which may hinder understanding of their functions.

Reviewer 2

Are you able to assess all statistics in the manuscript, including the appropriateness of statistical tests used? Yes, and I have assessed the statistics in my report.

Comments to author:

In this manuscript the authors show new aspects of the POL2A function, the catalytic subunit of DNA polymerase epsilon. The experiments were conducted with different mutant lines, all of them sharing strong phenotypes related to heterochromatin organization and gene silencing. By using deep-sequencing techniques they properly show that POL2A is needed for regulating non-CG methylation levels as well as H3K9 methylation. Perhaps, their most appealing discovery is that in order to regulate these two processes POL2A acts by somehow up-regulating CMT3 and by counteracting replicative stress. The authors also show that pol2a mutants show disrupted heterochromatin properties such as up-regulation of TEs and small chromocenters. These effects correlate with hypermethylation of non-CG sequences and increased deposition of H3K9me2. Interestingly, other heterochromatin features, such as H3K27me1 levels and H2A.W incorporation seem to be unaffected.

Overall the manuscript offers strong correlations for a possible role of POL2A in heterochromatin maintenance. Comments on the data provided are listed below.

1. Fig. 1C. All the point mutations shown in the gene model should be mentioned/described in the text, and explain why they are relevant for this study
2. It would be also informative to indicate where in the Pol2A molecule the novel point mutations reside and discuss this in relation with the function of this protein in DNA replication and heterochromatin maintenance.
3. H3K27me3 levels. Did the authors observe the presence of H3K4me3 in the target genes analyzed and affected in the mutants, and whether any change occurred relative to the wt? Could they be considered as having a bivalent nature?
4. Fig 1F and Fig S1G. In fact, the current Supplementary Figure is more informative than Fig 1F. I suggest moving this to Suppl information and S1G to the main text.
5. Fig S2D. Note that the TAIR accession codes are not correct for XRI1 and RAD51
6. Fig S3D. Error bars and statistical analysis are missing. Please check this throughout the manuscript.
7. Line 217. The authors show that mutation of pol2a in atxr56 reverts the over-replication phenotype (decondensed nuclei and ploidy profile) but not the HR and TE up-regulation. They conclude that POL2A is required for atxr56 induced chromatin over replication, however previous studies have shown that the over-replication phenotype is dependent on TE up-regulation, and that reverting the over replication phenotype does not revert the TE up regulation. I think the authors should include these observations before assessing a direct role of POL2A in atxr56 over-replication phenotypes. A more detailed discussion is needed in view of the transcription-replication conflicts reported for the atxr5/6 mutant.
8. Fig. 3B. Check order of panels in the legend.
9. Fig. 4C and S5J. I guess it should be pol2a-12.
10. Line 309. I guess is CAG instead of CAT.
11. Line 318. "...fas2-4 (fig S6E-F), there was no significant increase in the number of chromocenters per nucleus, differing from pol2a (fig 3, fig S6E). ..." What is the statistical significance to support this observation? Any reason for using the fas2-4 mutant instead of the (stronger?) fas1?
12. Line 424. Fig. 6F should be 6G.
13. Line 457. "We found no evidence of DSB accumulation in pol2a-12, and exposing pol2a plants to HU did not dramatize CHG hypermethylation." This sentence is not clear enough. Since H2AX phosphorylation can be carried out by both kinases (Amiard et al., 2010), quantifying the transcript or protein levels of ATR and ATM in WT and pol2a-12 plants with or without HU treatment, (or experiments with atm,atr mutants) would help in providing support to this conclusion.
14. Any changes in the ploidy profiles after the HU treatment?
15. Why do the authors think LTR/Gypsy are over-represented as the most abundant TE superfamily to

be up-regulated in *pol2a* when it is known that LTR/Copia TEs are regulated by non-CG methylation and H3K9me2 methylation?

16. POL2A seems to be involved in a pathway for heterochromatin maintenance that goes through CHG methylation and H3K9me2, as opposed to *atxr5/6*, although both are needed to silence the same family of TEs (LTR/Gypsy). How do you explain this, when it has been shown that non-CG methylation and H3K9me2 does not change in *atxr5/6*.

Reviewer 3

Are you able to assess all statistics in the manuscript, including the appropriateness of statistical tests used? No, I do not feel adequately qualified to assess the statistics.

Comments to author:

In their study, Bourguet et al investigated the role of DNA polymerase epsilon (ϵ) in heterochromatin structure and transcriptional gene silencing in Arabidopsis. As background, they note that the evidence for molecular components (proteins/machineries) linking DNA replication to epigenetic inheritance is scarce. After mutagenising the classic L5 transgenic line (Morel (2000) *Curr Biol.*), the authors obtained three mutant alleles of POL2a that release spontaneous transcriptional silencing of the GUS transgene in L5. This catalytic subunit of DNA Pol ϵ has several names (POL2a/TIL1/ESD7/EMB142/EMB529/EMB2284), testifying to past work that mainly explored developmental biology and abscisic acid sensitivity in *pol2a* mutants. Those past efforts found that *pol2a* plants have embryo patterning defects, a delayed progression through the cell cycle, as well as early flowering. Here, the authors characterised the mutant *anx2/pol2a-12* using an impressive number of genome-wide approaches.

DNA Pol ϵ mediates most leading strand elongation during DNA replication, so null homozygotes are embryo-lethal in Arabidopsis (Meinke (2019) *New Phytologist*). Because *anx2/pol2a-12* is a strong point mutant (but not null) it is a valuable tool for exploring links between DNA replication and epigenetic changes throughout the plant life cycle. Past reports linked POL2a to chromatin-mediated cellular memory, H3K27me3, H3K4me3 and H3KAc changes at specific genes in Arabidopsis, along with silencing of endogenous repeats (TSI, transcriptionally silent information)(Del Olmo (2009) *Plant J*, Yin (2009) *Plant Cell*).

The novel contributions of Bourguet are their genome wide analyses of DNA methylation, TE silencing/activation, chromatin states/condensation in *pol2a* mutants and how these phenomena modulate sensitivity to the DNA replication inhibitor/DNA repair inhibitor hydroxyurea. The study was carefully executed, featuring coherently reasoned data mining and displays that yield conceptual advances from precise quantitative results. Taken together, the authors show that TE control mediated by DNA Pol ϵ is similar to that mediated by ATXR5/6, with these sites corresponding to H3K27me1 peaks detected in chromatin immunoprecipitation (ChIP). Highly condensed, DAPI-stainable regions of chromatin (called chromocenters in Arabidopsis) disperse in the *pol2a-12* mutant, multiplying the per-nucleus number while reducing individual chromocenter area. The authors identified a series of TEs that show transcript accumulation in the *pol2a-8*, *pol2a-10* and *pol2a-12* mutants relative to wild type controls. All these experiments are well-controlled and backed by robust transcriptome data.

Global CHG methylation increases were observed in several DNA Pol ϵ mutant alleles (*pol2a-8*, *pol2a-10*, *pol2a-12*) compared to matched controls (L5 or WT) obtained from independent genetic screens. To explain this ultra-reproducible and specific CHG effect, the authors note slightly increased CMT3 and SUVH4 gene expression in *pol2a* plants. CMT3 and SUVH4 proteins maintain CHG methylation in plants, which could account for the increased CHG methylation (lines 367-369). This CMT3 gene

overexpression is not very obvious, though, as quantified via RT-PCR (figure S9L). Moreover, *atx5/6*, *pold2*, and *mail1* mutants show increases in CHG methylation that are *_not_* correlated with changes in *CMT3* gene expression. Based on this evidence the authors forward a second (more tenable) hypothesis, that CHG hypermethylation in *pol2a* mutants and other replisome-deficient plants is caused by DNA replication defects or DNA damage (lines 400-402). Hydroxyurea treatment data elegantly support the latter hypothesis (figure 6). Occam's razor suggests that the more parsimonious model should be favoured if two models make the same prediction. Implicating minor *CMT3* gene expression variation in major CHG methylation changes seems ad hoc: this line of reasoning could be de-emphasised to improve the manuscript.

In addition, the authors conclude that *POL2A* does not seem to promote accumulation of heterochromatic marks such as H3K27me1, H3K9me2 or H2A.W, but instead mediates aggregation of heterochromatic domains. Although topologically associating domains in mammals are not precisely equivalent to those seen in plant chromatin organization, future studies of DNA Pol ϵ could potentially reveal a role for this enzyme in higher-order, heterochromatin organization across diverse organisms. Yeast DNA Pol ϵ was previously implicated in transcriptional silencing of rDNA, silent mating-type loci and telomeres. The work of Bourguet et al. suggests that a (partially) conserved mechanism for DNA Pol ϵ -mediate silencing may exist (especially at rDNA), though they should cite the pre-2016 yeast genetics more thoroughly (e.g., Ehrenhofer-Murray (1999) *Genetics*, Smith et al (1999) *MCB*, Iida and Araki (2004) *MCB*).

In summary, the authors show that DNA Pol ϵ prevents CHG hypermethylation of TEs while mediating TE and gene silencing. They find that this CHG hypermethylation is observed in other mutants affecting replisome factors, such as *ATXR5/6*. The precise manner in which DNA Pol ϵ / replisome deficiencies cause CHG hypermethylation is not fully deciphered here, but the disaggregation of heterochromatin could explain the observed TE activation in *pol2a-12*. Publication of the authors study will be valuable because it raises novel connections between TE silencing, epigenetics and DNA repair. The role of this highly conserved DNA polymerase enzyme in TE repression will certainly be worth investigating in other organisms. For all these reasons, and in light of the elegantly presented genomic datasets, I recommend this study for publication in *Genome Biology* following minor revisions and corrections to the manuscript.

Queries and corrections:

1. lines 996, 1087 and perhaps elsewhere - "punctual mutations" would be "point mutations"
2. lines 1155 - "Differentially methylation positions (DMPs)" could be "Differentially methylated positions (DMPs)".
3. figure 1C - it would be helpful to also show the overall protein domain structure along with these mutations, like the diagrams in Pedroza-Garcia et al. (2019) *Int J Mol Sci.* or Del Olmo (2009) *Plant J.*
4. figure 1C - the *pol2a-4/til1-4 G3927A (G469R)* and *pol2a-4/til1-4 G5005A (intron)* labels are confusing. Does *til1-4* correspond to *_two_* closely spaced EMS mutations? If so, this should be mentioned. Or, is one of these actually *til1-3*? If so, it should have a different *pol2a* allele identifier too.
5. figure 4C, the grey title bar reads "*pol2a-6* upregulated TEs". I gather this should be "*pol2a-12* upregulated TEs".
6. figure S5I, there is a typo in the x-axis label "rmethylation"

POINT-BY-POINT RESPONSE TO REVIEWER'S COMMENTS

Reviewer #1: Bourguet et al. reported the isolation of several mutant alleles of DNA Pol ϵ through genetic screen for mutants that reactivate the GUS expression in Arabidopsis L5 line. They characterized the profiling of H3K27me3, H3K27me1, H3K9me2, DNA methylation, mRNA and small RNA-seq in WT, pol2a and related double or triple mutants as well as under HU treatments. Based on these analysis, the authors try to explore the function of DNA Pol ϵ in heterochromatin maintenance in Arabidopsis. Please see some major concerns on the manuscript.

1. The authors generated a lot of sequencing data in this study, but the details of those datasets are missing, including data quality and correlation between replicates, etc. The authors should have a basic description and a comprehensive table with these details in the manuscript.

A basic description of sequencing data has been included in the Methods section and a comprehensive table with sequencing read details is now provided in the new supplementary table 4. Correlation between RNA-seq and ChIP-seq replicates is shown in the new supplementary figure S10.

2. Line 180-184. The statement "...suggests that about half of PCG upregulation in pol2a-12 likely results from impaired PRC2-mediated H3K27me3 deposition..." might overstate the importance of H3K27me3. How about other types of histone modifications except for H3K27me3? I would rather say "...impaired PRC2-mediated H3K27me3 deposition might contribute to the about half of PCG upregulation in pol2a-12...", or "they are correlated". Similar statement in L197 needs to be considered.

POL2A physically interacts with PRC2 components such as CLF, EMF2 and MSI1, and it is required for both proper recruitment of CLF and EMF2 and proper H3K27me3 levels at the H3K27me3-regulated FT and SOC1 genes (del Olmo, 2016, Nucleic Acids Res). We find that about half of the PCGs upregulated in pol2a are associated with H3K27me3 and that these PCGs show slightly decreased H3K27me3 levels in pol2a (fig S2), suggesting similar regulation as at FT and SOC1.

Considering the reviewer's comment, we further investigated how loss of H3K27me3 impacts on the expression of the pol2a-upregulated genes associated with H3K27me3. Using published RNA-seq data (GSE67322), we find that these genes are upregulated in the curly leaf / swinger (clf swn) H3K27me3 methyltransferase mutant. These data have been added to the revised manuscript (see revised Fig S2D, and main text line 214-216).

Although we believe these data altogether suggest that impaired H3K27me3 deposition contribute to transcriptional upregulation at a set of H3K27me3-associated genes in pol2a, we agree with the reviewer that we cannot rule out a potential contribution of other histone modifications in the regulation of these genes. Following reviewer's suggestion, we toned down the corresponding statements about the importance of PRC2-mediated H3K27me3 deposition in pol2a-induced gene upregulation (see lines 216-220 and lines 249-251 of the revised manuscript).

In addition, the authors need to provide evidence to support that these upregulated PCG are not randomly overlapped with H3K27me3 mark.

We compared the proportion of PCGs associated with H3K27me3 in pol2a-12 upregulated genes (249 / 555, 44.9%) with that of H3K27me3-associated PCGs in the whole genome (8845 / 27206, 32.5%). Hypergeometric test indicates that PCGs upregulated in pol2a-12 are not randomly overlapped with the H3K27me3 mark, but significantly more associated with this mark ($P = 5.96e-10$). The figure S2B has been modified to illustrate this overlap.

3. Line 218: More data would be required to conclude that "POL2A is required for atxr5/6-induced heterochromatin overreplication", especially with several other negative correlations.

We used our BS-seq datasets to compute and compare the read ratio of atxr5/6, pol2a-12 and pol2a12 atxr5/6 over their respective wild types. We previously showed that this quantification can be used as a good proxy of DNA amount in the sequenced genotype (see Rigal et al. 2016 PNAS). This analysis confirms that heterochromatin is over-replicated in atxr5/6 but not in atxr5/6 pol2a-12, reinforcing the conclusion that POL2A is required for atxr5/6-induced heterochromatin over-replication. We added this data in the revised figure S3D.

4. The authors try to test the relationship between POL2A and other genes involving heterochromatin silencing across a couple of paragraphs. However, it seems that POL2A exerts its functions independently of these examined genes/proteins, like ATXR5/6, H2A.W and FAS2. These negative results provide limited knowledge for understanding the importance of POL2A as a central coordinator of heterochromatin structure and function. *Mutants for H3K27me1 (mediated by ATXR5/6) and H2A.W show altered heterochromatin structure (Jacob et al 2009 Nature, Yelagandula et al 2014 Cell). Strong DNA methylation mutants such as met1 and ddm1 also severely affect heterochromatin structure and function (Soppe et al. 2002 EMBO J). However, all data available to date indicate that DNA methylation, H3K27me1 and H2A.W are independent heterochromatin features that ensure its maintenance through independent routes (Jacob et al 2009 Nature, Yelagandula et al 2014 Cell, Ma et al 2018 Dev Cell, Mathieu et al 2005 EMBO J, Stroud et al 2013 Cell). Our data provide evidence that heterochromatin structure (chromocenters organization) and function (TE silencing) are compromised in pol2a mutants, and this occurs without reduction of DNA methylation (rather DNA hypermethylation) and changes in H2A.W association. Additionally, our mutant combination analyses suggest that POL2A regulates TE silencing and chromocenter organization independently of H3K27me1. Thus, our data suggest that POL2A is involved in yet another pathway that contributes to robustly maintain heterochromatin, in parallel to the other known pathways involving H3K27me1, DNA methylation or H2A.W. We believe our results provide strong evidence for the importance of POL2A in heterochromatin maintenance. However, we acknowledge that the expression "central coordinator" in the title is misleading as our data do not provide indication of a possible aspect of coordination between structure and function. To better convey our findings, we modified the title of the revised manuscript for "DNA polymerase epsilon is required for heterochromatin maintenance in Arabidopsis"*

Further, some data, such as sRNA-seq presented, could not well support its role of POL2A in heterochromatin structure and function.

Mutations in some genes such as MORC1 and MORC6 (Moissiard et al. 2012 Science), SMC4 (Wang et al. 2017 Genes Dev.), or MAIL1 (Ikeda et al. 2017 Nat Commun) alter heterochromatin organization without reducing sRNA levels. Reciprocally, mutations in factors required for siRNA biogenesis such as NRPD1, RDR2 and DCL3 have little consequence on pericentromere condensation (Pontes et al. 2009 Mol. Plant). Therefore, that POL2A does not promote siRNA biogenesis does not oppose to its important role in heterochromatin structure and function.

Some minor concerns:

1. A complementation test using genomic fragment containing wild-type DNA Pol ϵ gene to rescue the phenotype would provide solid genetic evidence for the study.

Introduction of a genomic fragment encompassing the wild-type POL2A gene was shown to rescue pol2a-10-associated developmental defects (del Olmo et al. 2010 Plant J.). Since we found non-CG DNA hypermethylation and release of TE silencing to consistently occur in several independent pol2a mutant alleles, making the possibility for a secondary mutation causing these molecular phenotypes very unlikely, we do not think a complementation test is necessary for the current study.

2. According to above words, I think the title of the manuscript overstates the findings and the statement "DNA polymerase epsilon is a central coordinator" is far-fetched.

As explained above, the title has been modified for "DNA polymerase epsilon is required for heterochromatin maintenance in Arabidopsis".

3. Line 59-60, the authors claims that "euchromatin and heterochromatin are two main different organization states..." is oversimplified and could not match the conclusions from recent published a couple of plants ENCODE papers. At least 4-5 different types of chromatin states were reported.

We fully agree with the reviewer that this is likely an oversimplification and that several (many) chromatin organization subtypes may be distinguished based on various criteria, for instance association with specific combinations of histone marks.

As our intention in the sentence of the introduction the reviewer refers to is to introduce the big difference between the two major states of chromatin that we consider throughout our study, we would prefer to leave it as such not to unnecessarily complexify the introduction. Nonetheless, we modified our wording to emphasize that this is a simplified view (see line 65 of the revised manuscript).

4. Line 164. Please provide numbers for total DEGs.

Done (see line 196 of the revised manuscript).

5. L187-194: please add the percentages also for the upregulated PCGs involved in replicative stress and cell cycle.

These percentages are provided in the revised manuscript (see line 248).

6. Lots of genes are not given a full name at the first time appearance, which may hinder understanding of their functions.

Thank you for bringing this to our attention. This is now corrected in the revised manuscript.

Reviewer #2:

In this manuscript the authors show new aspects of the POL2A function, the catalytic subunit of DNA polymerase epsilon. The experiments were conducted with different mutant lines, all of them sharing strong phenotypes related to heterochromatin organization and gene silencing. By using deep-sequencing techniques they properly show that POL2A is needed for regulating non-CG methylation levels as well as H3K9 methylation. Perhaps, their most appealing discovery is that in order to regulate these two processes POL2A acts by somehow up-regulating CMT3 and by counteracting replicative stress. The authors also show that *pol2a* mutants show disrupted heterochromatin properties such as up-regulation of TEs and small chromocenters. These effects correlate with hypermethylation of non-CG sequences and increased deposition of H3K9me2. Interestingly, other heterochromatin features, such as H3K27me1 levels and H2A.W incorporation seem to be unaffected. Overall the manuscript offers strong correlations for a possible role of POL2A in heterochromatin maintenance. Comments on the data provided are listed below.

1. Fig. 1C. All the point mutations shown in the gene model should be mentioned/described in the text, and explain why they are relevant for this study

*We included the *til1-4* mutation in the gene model to show the consistency of the new nomenclature of *pol2a* mutant alleles proposed in our study with the original nomenclature of *til1* mutants. All other mutations are shown since they are first reported in the current study (*pol2a-12*, *pol2a-13*, *pol2a-14*) or used in our experiments (*pol2a-8*, *pol2a-10*). We now describe *pol2a* mutations in the legend of figure 1 to keep the main text short.*

2. It would be also informative to indicate where in the Pol2A molecule the novel point mutations reside and discuss this in relation with the function of this protein in DNA replication and heterochromatin maintenance.

*In light of this comment (and to point #3 of reviewer #3), we added a protein model to figure 1C wherein POL2A domains are indicated alongside *pol2a* point mutations used in this study. Interestingly, *pol2a-8*, *pol2a-10* and *pol2a-12* mutations all induce TE silencing defects and CHG hypermethylation and all affect the catalytic domain of POL2A involved in DNA replication. This supports the notion that replicative defects contribute to loss of TE silencing and CHG hypermethylation in the analyzed *pol2a* mutants and it is now mentioned in the result (lines 181-83; 470-71) and discussion (line 524-25) sections. We thank reviewers #2 and #3 for their insightful suggestions.*

3. H3K27me3 levels. Did the authors observe the presence of H3K4me3 in the target genes analyzed and affected in the mutants, and whether any change occurred relative to the wt? Could they be considered as having a bivalent nature?

*We analyzed the association of *pol2a-12* upregulated genes with H3K4me3 and found that these genes are rather depleted in this mark (see figure below, left panel), whereas they show clear enrichment in H3K27me3 (see figure below, right panel). Thus, these genes do not show an H3K4/K27me3 bivalent chromatin signature. This new observation has been added to the manuscript (lines 219-221) and to figure S2.*

4. Fig 1F and Fig S1G. In fact, the current Supplementary Figure is more informative than Fig 1F. I suggest moving this to Suppl information and S1G to the main text.

Following the reviewer's request, we swapped Fig 1F and Fig S1G in the revised manuscript.

5. Fig S2D. Note that the TAIR accession codes are not correct for XRI1 and RAD51

Thank you, this mistake has been corrected in the revised manuscript.

6. Fig S3D. Error bars and statistical analysis are missing. Please check this throughout the manuscript.

Fig S3D shows representative flow cytometry profiles, so we cannot provide error bars here. We included additional replicates in the data presented in Fig S3E and added the corresponding error bars and statistical analysis.

7. Line 217. The authors show that mutation of pol2a in atrx56 reverts the over-replication phenotype (decondensed nuclei and ploidy profile) but not the HR and TE up-regulation. They conclude that POL2A is required for atrx56 induced chromatin over replication, however previous studies have shown that the over-replication phenotype is dependent on TE up-regulation, and that reverting the over replication phenotype does not revert the TE up regulation. I think the authors should include these observations before assessing a direct role of POL2A in atrx56 over-replication phenotypes. A more detailed discussion is needed in view of the transcription-replication conflicts reported for the atrx5/6 mutant.

Indeed, in a screen for mutants that suppress the atrx5/6 phenotype, Hale et al. (2016 Plos Genet.) found that every suppressor they isolated reduced both the transposon derepression phenotype and the overreplication phenotype of atrx5/6. Based on their data, Hale et al. (2016 Plos Genet.) favored a model wherein atrx5/6 mutations cause transcriptional defects that indirectly cause the over-replication phenotype. However, as explicitly stated by the authors themselves, a model wherein atrx5/6 mutations affect replication and silencing independently could not be ruled out (Hale et al. 2016 Plos Genet.).

Our data show that pol2a-12 suppresses DNA over-replication but not the transcriptional defects caused by atrx5/6, which is consistent with the model wherein atrx5/6 mutations may affect replication and transcription independently. In support of this interpretation, a previous study reported that mutations in DNA methyltransferases similarly suppress the over-replication caused by atrx5/6, but not the TE transcriptional derepression (Stroud et al. 2012 Plos Genet.). We also note that the reported absence of DNA over-replication in several silencing mutants (Stroud et al. 2012 Plos Genet.) indicates that loss of TE silencing is not sufficient per se to trigger DNA over-replication. This is now mentioned in the corresponding result section (lines 291-93).

8. Fig. 3B. Check order of panels in the legend.

Thank you. This is now corrected in the revised manuscript.

9. Fig. 4C and S5J. I guess it should be pol2a-12.

This typo is corrected in the revised manuscript.

10. Line 309. I guess is CAG instead of CAT.

Thank you. This is now corrected in the revised manuscript.

11. Line 318. "...fas2-4 (fig S6E-F), there was no significant increase in the number of chromocenters per nucleus, differing from pol2a (fig 3, fig S6E). ..." What is the statistical significance to support this observation? Any reason for using the fas2-4 mutant instead of the (stronger?) fas1?

We made a mistake generating the boxplots shown in fig S6E: the fas2 data is incorrect for relative chromocenter area and chromocenter per nucleus, while the whole plot is wrong for relative heterochromatic fraction. The figure has been updated with correct data and corresponding p-values. This corrected figure shows that the number of chromocenters per nucleus in fas2 is indeed not significantly different from the WT (p-value=0.231).

We used the fas2-4 mutant allele in our transcriptomic and genetic experiments because of the similarities between pol2a and fas2 mutant phenotypes and because the fas2-4 mutant allele was already known to suppress atrx5/6-induced heterochromatin over-replication (Jacob et al. Science 2014), like pol2a-12. For the sake of consistency, we also used this allele for cytological analyses.

12. Line 424. Fig. 6F should be 6G.

This is corrected in the revised manuscript.

13. Line 457. "We found no evidence of DSB accumulation in pol2a-12, and exposing pol2a plants to HU did not dramatize CHG hypermethylation." This sentence is not clear enough. Since H2AX phosphorylation can be carried out by both kinases (Amiard et al., 2010), quantifying the transcript or protein levels of ATR and ATM in WT and pol2a-12 plants with or without HU treatment, (or experiments with atm,atr mutants) would help in providing support to this conclusion.

We agree that the proposed experiments may help clarifying the respective contributions of ATR and ATM. However, since these two kinases regulate many other proteins, interpreting the results would be rather complex and we would like to save these experiments for potential future independent studies. As the respective contributions of ATR and/or ATM to the phenotypes of pol2a, atxr5/6 and fas2 mutants is not instrumental in interpreting of our data, we removed the related statements from the discussion.

14. Any changes in the ploidy profiles after the HU treatment?

We have not assessed whether HU treatment induces ploidy changes.

15. Why do the authors think LTR/Gypsy are over-represented as the most abundant TE superfamily to be up-regulated in pol2a when it is known that LTR/Copia TEs are regulated by non-CG methylation and H3K9me2 methylation?

LTR/Copia TEs are preferentially located along chromosome arms, while LTR/Gypsy elements are largely clustered within pericentromeric heterochromatin. LTR/Gypsy TEs are overrepresented in pol2a-activated TEs because pol2a mutations mostly destabilize silencing of pericentromeric regions.

16. POL2A seems to be involved in a pathway for heterochromatin maintenance that goes through CHG methylation and H3K9me2, as opposed to atxr5/6, although both are needed to silence the same family of TEs (LTR/Gypsy). How do you explain this, when it has been shown that non-CG methylation and H3K9me2 does not change in atxr5/6.

Generating new atxr5/6 BS-seq data and reanalyzing previously published datasets, we show in the current study that CHG DNA methylation and H3K9me2 levels are actually increased in atxr5/6 mutants (see FigS9 A-D). Therefore, transcriptional activation of LTR/Gypsy TEs is associated with increased levels of these two epigenetic marks in both pol2a and atxr5/6 mutants.

Reviewer #3: In their study, Bourguet et al investigated the role of DNA polymerase epsilon (ϵ) in heterochromatin structure and transcriptional gene silencing in Arabidopsis. As background, they note that the evidence for molecular components (proteins/machineries) linking DNA replication to epigenetic inheritance is scarce. After mutagenising the classic L5 transgenic line (Morel (2000) Curr Biol.), the authors obtained three mutant alleles of POL2a that release spontaneous transcriptional silencing of the GUS transgene in L5. This catalytic subunit of DNA Pol ϵ has several names (POL2a/TIL1/ESD7/EMB142/EMB529/EMB2284), testifying to past work that mainly explored developmental biology and abscisic acid sensitivity in pol2a mutants. Those past efforts found that pol2a plants have embryo patterning defects, a delayed progression through the cell cycle, as well as early flowering. Here, the authors characterised the mutant anx2/pol2a-12 using an impressive number of genome-wide approaches.

DNA Pol ϵ mediates most leading strand elongation during DNA replication, so null homozygotes are embryo-lethal in Arabidopsis (Meinke (2019) New Phytologist). Because anx2/pol2a-12 is a strong point mutant (but not null) it is a valuable tool for exploring links between DNA replication and epigenetic changes throughout the plant life cycle. Past reports linked POL2a to chromatin-mediated cellular memory, H3K27me3, H3K4me3 and H3KAc changes at specific genes in Arabidopsis, along with silencing of endogenous repeats (TSI, transcriptionally silent information)(Del Olmo (2009) Plant J, Yin (2009) Plant Cell).

The novel contributions of Bourguet are their genome wide analyses of DNA methylation, TE silencing/activation, chromatin states/condensation in pol2a mutants and how these phenomena modulate sensitivity to the DNA replication inhibitor/DNA repair inhibitor hydroxyurea. The study was carefully executed, featuring coherently reasoned data mining and displays that yield conceptual advances from precise quantitative results. Taken together, the authors show that TE control mediated by DNA Pol ϵ is similar to that mediated by ATXR5/6, with these sites corresponding to H3K27me1 peaks detected in chromatin immunoprecipitation (ChIP). Highly condensed, DAPI-stainable regions of chromatin (called chromocenters in Arabidopsis) disperse in the pol2a-12 mutant, multiplying the per-nucleus number while reducing individual chromocenter area. The authors identified a series of TEs that show transcript accumulation in the pol2a-8, pol2a-10 and pol2a-12 mutants relative to wild type controls. All these experiments are well-controlled and backed by robust transcriptome data.

Global CHG methylation increases were observed in several DNA Pol ϵ mutant alleles (pol2a-8, pol2a-10, pol2a-12) compared to matched controls (L5 or WT) obtained from independent genetic screens. To explain this ultra-reproducible and specific CHG effect, the authors note slightly increased CMT3 and SUVH4 gene expression in pol2a plants. CMT3 and SUVH4 proteins maintain CHG methylation in plants, which could account for the increased CHG methylation (lines 367-369). This CMT3 gene overexpression is not very obvious, though, as quantified via RT-PCR (figure S9L). Moreover, atxr5/6, pold2, and mail1 mutants show increases in CHG methylation that are not correlated with changes in CMT3 gene expression. Based on this evidence the authors forward a second (more tenable) hypothesis, that CHG hypermethylation in pol2a mutants and other replisome-deficient plants is caused by DNA replication defects or DNA damage (lines 400-402). Hydroxyurea treatment

data elegantly support the latter hypothesis (figure 6). Occam's razor suggests that the more parsimonious model should be favoured if two models make the same prediction. Implicating minor CMT3 gene expression variation in major CHG methylation changes seems ad hoc: this line of reasoning could be de-emphasised to improve the manuscript.

In addition, the authors conclude that POL2A does not seem to promote accumulation of heterochromatic marks such as H3K27me1, H3K9me2 or H2A.W, but instead mediates aggregation of heterochromatic domains. Although topologically associating domains in mammals are not precisely equivalent to those seen in plant chromatin organization, future studies of DNA Pol ϵ could potentially reveal a role for this enzyme in higher-order, heterochromatin organization across diverse organisms. Yeast DNA Pol ϵ was previously implicated in transcriptional silencing of rDNA, silent mating-type loci and telomeres. The work of Bourguet et al. suggests that a (partially) conserved mechanism for DNA Pol ϵ -mediate silencing may exist (especially at rDNA), though they should cite the pre-2016 yeast genetics more thoroughly (e.g., Ehrenhofer-Murray (1999) Genetics, Smith et al (1999) MCB, Iida and Araki (2004) MCB).

We now refer to this earlier work in yeast in the revised manuscript.

In summary, the authors show that DNA Pol ϵ prevents CHG hypermethylation of TEs while mediating TE and gene silencing. They find that this CHG hypermethylation is observed in other mutants affecting replisome factors, such as ATXR5/6. The precise manner in which DNA Pol ϵ / replisome deficiencies cause CHG hypermethylation is not fully deciphered here, but the disaggregation of heterochromatin could explain the observed TE activation in pol2a-12. Publication of the authors study will be valuable because it raises novel connections between TE silencing, epigenetics and DNA repair. The role of this highly conserved DNA polymerase enzyme in TE repression will certainly be worth investigating in other organisms. For all these reasons, and in light of the elegantly presented genomic datasets, I recommend this study for publication in Genome Biology following minor revisions and corrections to the manuscript.

We thank the reviewer for her/his thorough and very positive assessment of our work.

Queries and corrections:

1. lines 996, 1087 and perhaps elsewhere - "punctual mutations" would be "point mutations"

Thank you. This is corrected in the revised manuscript.

2. lines 1155 - "Differentially methylation positions (DMPs)" could be "Differentially methylated positions (DMPs).

We have corrected this typo.

3. figure 1C - it would be helpful to also show the overall protein domain structure along with these mutations, like the diagrams in Pedroza-Garcia et al. (2019) Int J Mol Sci. or Del Olmo (2009) Plant J.

Please see our response to the point #2 of reviewer #2. We added a protein model to figure 1C wherein POL2A domains are indicated alongside pol2a point mutations used in this study. Interestingly, pol2a-8, pol2a-10 and pol2a-12 mutations all induce TE silencing defects and CHG hypermethylation and all affect the catalytic domain of POL2A involved in DNA replication. This supports the notion that replicative defects contribute to loss of TE silencing and CHG hypermethylation in the analyzed pol2a mutants and it is now mentioned in the result (lines 181-183; 470-471) and discussion (line 524-5) sections. We thank reviewers #2 and #3 for their insightful suggestions.

4. figure 1C - the pol2a-4/til1-4 G3927A (G469R) and pol2a-4/til1-4 G5005A (intron) labels are confusing. Does til1-4 correspond to two closely spaced EMS mutations? If so, this should be mentioned. Or, is one of these actually til1-3? If so, it should have a different pol2a allele identifier too.

Indeed til1-4 mutants contain two point mutations in the POL2A gene -one in exon 12 and one in intron 14. Following the reviewer's request, this is now clearly mentioned in the revised legend of figure 1C.

5. figure 4C, the grey title bar reads "pol2a-6 upregulated TEs". I gather this should be "pol2a-12 upregulated TEs".

We have corrected this typo.

6. figure S5I, there is a typo in the x-axis label "r methylation"

This is corrected in the revised manuscript.

Second round of review

Reviewer 1

The authors have adequately addressed my comments. The revised manuscript is now greatly improved.

Reviewer 2

I appreciate the effort made by the authors to address the points that I listed in my report. They have satisfactorily answered most, if not all of them. Consequently I have no more concerns about the results and claims of this manuscript.